# CORESET SELECTION VIA REDUCIBLE LOSS IN CONTINUAL LEARNING

**Ruilin Tong**[1], **Yuhang Liu**[2], **Javen Qinfeng Shi**[2], **Dong Gong**[1]*
[1] School of Computer Science and Engineering, University of New South Wales
[2] Australian Institute for Machine Learning, The University of Adelaide
{ruilin.tong, dong.gong}@unsw.edu.au
{yuhang.liu01, javen.shi}@adelaide.edu.au

## ABSTRACT

Rehearsal-based continual learning (CL) aims to mitigate catastrophic forgetting by maintaining a subset of samples from previous tasks and replaying them. The rehearsal memory can be naturally constructed as a coreset, designed to form a compact subset that enables training with performance comparable to using the full dataset. The coreset selection task can be formulated as bilevel optimization that solves for the subset to minimize the outer objective of the learning task. Existing methods primarily rely on inefficient probabilistic sampling or local gradient-based scoring to approximate sample importance through an iterative process that can be susceptible to ambiguity or noise. Specifically, non-representative samples like ambiguous or noisy samples are difficult to learn and incur high loss values even when training on the full dataset. However, existing methods relying on local gradient tend to highlight these samples in an attempt to minimize the outer loss, leading to a suboptimal coreset. To enhance coreset selection, especially in CL where high-quality samples are essential, we propose a coreset selection method that measures sample importance using reducible loss (ReL) that quantifies the impact of adding a sample to model performance. By leveraging ReL and a process derived from bilevel optimization, we identify and retain samples that yield the highest performance gain. They are shown to be informative and representative. Furthermore, ReL requires only forward computation, making it significantly more efficient than previous methods. To better apply coreset selection in CL, we extend our method to address key challenges such as task interference, streaming data, and knowledge distillation. Experiments on data summarization and continual learning demonstrate the effectiveness and efficiency of our approach. Our code is available at https://github.com/RuilinTong/CSReL-Coreset-CL.

## 1 INTRODUCTION

Continual learning (CL) aims to learn novel knowledge from a non-stationary stream of data containing different tasks, while maintaining the learned knowledge (Ring, 1997; Rebuffi et al., 2017). Models tend to forget previously learned tasks when they are trained on new ones, a phenomenon known as catastrophic forgetting (McCloskey & Cohen, 1989). A straightforward and effective way for countering forgetting is experience replay (ER) (Buzzega et al., 2020; Lopez-Paz & Ranzato, 2017; Chaudhry et al., 2019), which maintains a small rehearsal memory to store a subset of previous data and replay it during training on new tasks to mitigate forgetting. It is also convenient to incorporate ER with other types of CL methods, e.g. parameter isolation-based methods (Yan et al., 2022; Wang et al., 2022a) and regularization-based methods (Kirkpatrick et al., 2017). Early methods for ER often selected old data randomly (Vitter, 1985; Lopez-Paz & Ranzato, 2017; Chaudhry et al., 2018b; Buzzega et al., 2020; Riemer et al., 2018). However, this indiscriminate approach may not yield optimal results, as selecting an informative subset of data for memory storage is crucial for the effectiveness of experience replay.

---

*D. Gong is the corresponding author.

A natural way to enhance ER is to select a coreset for memory, which is a small subset of the dataset designed to allow a model trained on it to achieve performance comparable to one trained on the full dataset. The coreset selection can be formulated as a bilevel optimization problem that solves for the subset over the full dataset to minimize the outer training objective. Greedy Coreset (Borsos et al., 2020) iteratively adds samples into the coreset through an approximation of the implicit gradient as an indicator of the sample importance and a greedy solver of the bilevel problem. PBCS (Zhou et al., 2022b) formulates the coreset with probabilistic masks on samples and solves the coreset selection problem from a global perspective, which relies on inefficient sampling process. BCSR (Hao et al., 2024) learns probabilities while preserving the nested structure of bilevel optimization.

Bilevel coreset selection methods usually treat all samples equally in the outer loss. However, non-representative samples, such as ambiguous or noisy ones that are common in real-world data, are often not well learned, even when training on the full dataset. This suggests that these samples are not valuable to be represented by coreset. In the bilevel selection framework, these samples heavily impact the outer loss due to their high loss values. In methods that rely on an incremental coreset expansion process with approximate implicit gradients (Borsos et al., 2020), these samples are often assigned high scores for reducing the outer loss and tend to be selected into the coreset. Selecting ambiguous or noisy samples hinders the improvement of model performance after training on coreset. Additionally, previous bilevel coreset selection methods are computationally expensive. The computation of inverse Hessian matrix is computationally expensive (Borsos et al., 2020; Hao et al., 2024). PBCS (Zhou et al., 2022b) relying on policy gradient requires a large number of sampling iterations, and reducing sampling numbers degrades performance significantly.

To enhance coreset selection, especially in CL where high-quality samples are essential, we propose a coreset selection method that measures sample importance using reducible loss (ReL) (Mindermann et al., 2022), named CSReL. We study how the model's performance changes when a sample is added to the training dataset and quantify this based on ReL. The CSReL can effectively highlight an informative sample while excluding ambiguous or noisy ones according to the model's status. We assert that samples with high performance gains are representative and contribute relatively new knowledge compared to the already selected samples. Our method integrates the greedy incremental framework from bilevel optimization with ReL. In the incremental process, ReL is computed as the difference between the losses on the remaining samples computed by a holdout model trained on the full dataset, and a current model trained on the selected subset. Samples with the highest ReL are selected into the coreset for maximizing performance gain. ReL only requires forward computation (Mindermann et al., 2022), without the need for backward computation, making it significantly more efficient than other methods. To better adapt coreset selection to rehearsal-based CL, we extend our method to address the unique challenges of CL, including task interference, streaming data, and knowledge distillation. Our main contributions are:

- We address the issue of selecting ambiguous or noisy samples in previous bilevel coreset selection methods by proposing a coreset selection approach based on ReL reflecting the performance gain. We show that samples with high performance gain are both representative and informative, and our selection criterion effectively prevents the inclusion of ambiguous or noisy samples.

- We propose an efficient coreset selection approach based on ReL, which is well-suited to be extended for addressing the unique challenges of CL: 1) Reducing task interference, 2) Selecting coresets from streaming data, and 3) Selecting coresets for knowledge distillation.

- We demonstrate the effectiveness and efficiency of our method through extensive experiments on data summarization and CL tasks. Additionally, we show the effectiveness and compatibility of our method by enhancing the performance of existing CL methods.

## 2 RELATED WORK

**Coreset selection.** Coreset selection aims to select the most informative subset from full dataset. Previous coreset selection methods are designed for $k$-means (Feldman & Langberg, 2011), Gaussian mixture model (Lucic et al., 2018), logistic regression (Huggins et al., 2016) and Bayesian inference (Campbell & Broderick, 2019). These methods are only suitable for traditional methods, while cannot be applied in deep neural networks. Greedy Coreset (Borsos et al., 2020) extended coreset selection to deep neural networks by formulating coreset selection problem as a bilevel optimization problem. PBCS (Zhou et al., 2022b) selects globally with probabilistic masks. BCSR (Hao et al., 2024) considers nested nature based on PBCS (Zhou et al., 2022b). Previous bilevel selection methods may select ambiguous or noisy samples in an effort to reduce the outer loss. Additionally,

these methods are computationally expensive. Our coreset selection approach effectively prevents the selection of ambiguous and noisy samples while being more efficient.

**Continual learning.** Continual learning aims to adapt learning agent to sequence of tasks, and previous tasks is not available once learnt, including regularization-based methods (Kirkpatrick et al., 2017; Zenke et al., 2017; Chaudhry et al., 2018a; Ritter et al., 2018; Li & Hoiem, 2017; Nguyen et al., 2017; Ebrahimi et al., 2019; Yan et al., 2022; Jha et al., 2023), parameter isolation based methods (Yan et al., 2021; Wang et al., 2022a; Zhou et al., 2022a; Jin et al., 2023; Ostapenko et al., 2021; Wang et al., 2024) and rehearsal-based methods (Lopez-Paz & Ranzato, 2017; Rebuffi et al., 2017; Chaudhry et al., 2018b; Riemer et al., 2018; Aljundi et al., 2019c; Chaudhry et al., 2019; Borsos et al., 2020; Zhou et al., 2022b; Yoon et al., 2021; Aljundi et al., 2019a; Isele & Cosgun, 2018; Buzzega et al., 2021; Caccia et al., 2021). In this work, we focus on rehearsal-based methods and sample selection for maintaining high-quality memory.

**Coreset for continual learning.** Selecting samples for memory plays a crucial role in the performance of rehearsal-based CL. Previous works, such as Aljundi et al. (2019c); Sun et al. (2022b); Bang et al. (2021); Wiewel & Yang (2021); Hurtado et al. (2023), are primarily based on heuristic insights. Wang et al. (2022b) proposes compressing memory data to store more samples, while OCS (Yoon et al., 2021) and GCR (Tiwari et al., 2022) select coresets through gradient matching. Greedy Coreset (Borsos et al., 2020), BCSR (Hao et al., 2024), and PBCS (Zhou et al., 2022b) apply coreset selection directly to single-task datasets for memory. We investigate effective coreset selection method and the application of it for CL, while considering task interference, streaming scenarios, and knowledge distillation, in CL.

## 3 CORESET SELECTION WITH REDUCIBLE LOSS

### 3.1 BILEVEL OPTIMIZATION FOR CORESET SELECTION

Coreset selection aims to select a subset from a given dataset $\mathcal{D}$ so that the model trained on coreset could achieve comparable performance as the model trained on $\mathcal{D}$. The selection can be achieved via weighing the samples. Let loss function on the $i$-th sample be $\ell(x_i, y_i; \boldsymbol{\theta})$ with $\boldsymbol{\theta}$ being model parameters. Coreset selection can be formulated a bilevel optimization problem (Borsos et al., 2020):

$$\min_{\mathbf{w} \in \mathbb{R}_+^{|\mathcal{D}|}, \|\mathbf{w}\|_0 \leq m} \mathcal{L}(\boldsymbol{\theta}^*(\mathbf{w})) \coloneqq \sum_{i=1}^{|\mathcal{D}|} \ell(x_i, y_i; \boldsymbol{\theta}^*(\mathbf{w})),$$

$$\text{s.t.} \quad \boldsymbol{\theta}^*(\mathbf{w}) \in \arg\min_{\boldsymbol{\theta}} \hat{\mathcal{L}}(\boldsymbol{\theta}) \coloneqq \sum_{i=1}^{|\mathcal{D}|} w_i \ell(x_i, y_i; \boldsymbol{\theta})), \tag{1}$$

where $w_i$ is selection weight for sample $(x_i, y_i)$, $\mathbf{w}$ is corresponding selection vector, $\boldsymbol{\theta}^*(\mathbf{w})$ is solution for inner optimization problem, $|\mathcal{D}|$ denotes the number of samples in $\mathcal{D}$. Coreset size is constrained by $\|\mathbf{w}\|_0 \leq m$.

Although the problem in Eq. (1) is known to be NP-hard, it can be heuristically solved using first-order approaches (Borsos et al., 2020). The first-order optimality condition for the minimizer $\boldsymbol{\theta}^*(\mathbf{w})$ of $\hat{\mathcal{L}}(\boldsymbol{\theta})$ implies that the gradient of $\hat{\mathcal{L}}(\boldsymbol{\theta})$ with respect to (w.r.t.) $\boldsymbol{\theta}$ evaluated at $\boldsymbol{\theta}^*(\mathbf{w})$ must vanish, i.e., $\frac{\partial \hat{\mathcal{L}}(\boldsymbol{\theta}^*(\mathbf{w}))}{\partial \boldsymbol{\theta}^*(\mathbf{w})} = 0$. This reflects that, at the optimal results, the gradient of the loss w.r.t. $\boldsymbol{\theta}$ vanishes. Relying on the implicit function theorem, we can obtain the gradient of $\boldsymbol{\theta}^*(\mathbf{w})$ w.r.t. $\mathbf{w}$, i.e., the implicit gradient, enabling us to compute the gradient of outer loss function $\mathcal{L}(\boldsymbol{\theta}^*(\mathbf{w}))$ w.r.t. the selection weights $\mathbf{w}$. Focusing on the importance of each sample, the gradient w.r.t. each $w_i$ is derived as follows:

$$\nabla_{w_i} \mathcal{L}(\boldsymbol{\theta}^*(\mathbf{w})) = -\nabla_{\boldsymbol{\theta}} \mathcal{L}(\boldsymbol{\theta}^*(\mathbf{w}))^T \mathbf{H}^{-1} \nabla_{\boldsymbol{\theta}} \ell(x_i, y_i; \boldsymbol{\theta}^*(\mathbf{w})), \tag{2}$$

where $\mathbf{H}$ denotes Hessian matrix of $\hat{\mathcal{L}}(\boldsymbol{\theta}^*(\mathbf{w}))$. The irrelevant items are dropped in Eq. (2); and more details are in Appendix D.

To solve for $\mathbf{w}$ with the coreset size constraint, Greedy Coreset (Borsos et al., 2020) integrates the first-order method into a cone constrained generalized matching pursuit approach (Locatello et al., 2017). This method iteratively selects samples into the coreset before reaching the size constraint. At each iteration step, given a $\mathcal{S}$, new sample with the largest negative implicit gradient among the remaining samples is selected. Considering that $\boldsymbol{\theta}^*(\mathbf{w})$ is trained on $\mathcal{S}$ in the greedy process, we use a concise notation $\boldsymbol{\theta}_{\mathcal{S}}$ in place of $\boldsymbol{\theta}^*(\mathbf{w})$. Through the connection between the bilevel optimization

task and influence functions (Koh & Liang, 2017), Eq. (2) can be rewritten as

$$\nabla_{w_i} \mathcal{L}(\boldsymbol{\theta}_{\mathcal{S}}) = -\nabla_{\boldsymbol{\theta}} \mathcal{L}(\boldsymbol{\theta}_{\mathcal{S}})^T \mathbf{H}^{-1} \ell(x_i, y_i; \boldsymbol{\theta}_{\mathcal{S}}) = \nabla_{\boldsymbol{\theta}} \mathcal{L}(\boldsymbol{\theta}_{\mathcal{S}})^T \nabla_{w_i} \boldsymbol{\theta}_{\mathcal{S}}, \tag{3}$$

where $\nabla_{w_i} \boldsymbol{\theta}_{\mathcal{S}}$ represents the optimal parameter change when $(x_i, y_i)$ is added to $\mathcal{S}$ and the model is retrained (Borsos et al., 2020). Eq. (3) shows that the gradient of $w_i$ can be expressed as the inner product between the outer loss gradient $\nabla_{\boldsymbol{\theta}} \mathcal{L}(\boldsymbol{\theta}_{\mathcal{S}})$ and the optimal parameter change $\nabla_{w_i} \boldsymbol{\theta}_{\mathcal{S}}$.

Despite the above progress, non-representative samples, such as ambiguous or noisy samples, may exhibit large loss values, contributing significantly to $\mathcal{L}(\boldsymbol{\theta}_{\mathcal{S}})$. Including such samples in the coreset can reduce $\mathcal{L}(\boldsymbol{\theta}_{\mathcal{S}})$, leading to a negative inner product in Eq. (3) and significant parameters changes. According to the gradient-based sample importance score as in Eq. (2) and (3), a sample causing large changes in parameters and gradients can have a high likelihood of being selected into $\mathcal{S}$, which may not be helpful for optimizing the real learning objective. The related effect has been observed in the coreset selection process of Greedy Coreset (Borsos et al., 2020), with detailed experiments and results provided in Appendix H.

### 3.2 MAXIMIZE PERFORMANCE GAIN IN CORESET SELECTION WITH ReL

Selecting ambiguous or noisy samples into coreset can lead to ineffectiveness in ensuring the performance of the model trained on it or even degrading it, which can be more crucial in CL with only restricted memory size for reply. To address these issues, we propose a coreset selection method by incorporating the reducible loss (Mindermann et al., 2022) into the greedy incremental selection scheme derived from bilevel optimization. To alleviate the issue caused by gradient-based score, we try to directly capture the reduction of outer objective in Eq. (1) as the selection objective. It aims to select samples that lead to the maximum reduction in the outer objective when added to the coreset, $\mathcal{S}$, and can also been seen as rephrase the gradient-based score relying on Eq. (2) and (3) with the original objectives. Specifically, when adding $(x_i, y_i)$ into $\mathcal{S}$, we consider the following gain:

$$G_i = \log p(\mathbf{y}|\mathbf{x}; \mathcal{S} \cup (x_i, y_i)) - \log p(\mathbf{y}|\mathbf{x}; \mathcal{S}), \tag{4}$$

where $(\mathbf{x}, \mathbf{y})$ denotes all samples in $\mathcal{D}$ and $(x_i, y_i) \in \mathcal{D} \backslash \mathcal{S}$. Eq. (4) is not tractable to $(x_i, y_i)$ and selecting samples with maximum performance gain requires training models on every $\mathcal{S} \cup (x_i, y_i)$, which is impractical. Following Mindermann et al. (2022), we apply Bayes rule and conditional independence to make computation of $G_i$ tractable to $(x_i, y_i)$ as

$$\log p(\mathbf{y}|\mathbf{x}; \mathcal{S} \cup (x_i, y_i)) = \log \frac{p(y_i|x_i; \mathcal{D} \cup \mathcal{S}) \, p(\mathbf{y}|\mathbf{x}, x_i; \mathcal{S})}{p(y_i|x_i, \mathbf{x}; \mathcal{S})} = \log \frac{p(y_i|x_i; \mathcal{D} \cup \mathcal{S}) \, p(\mathbf{y}|\mathbf{x}; \mathcal{S})}{p(y_i|x_i; \mathcal{S})}$$

$$\log p(\mathbf{y}|\mathbf{x}; \mathcal{S} \cup (x_i, y_i)) - \log p(\mathbf{y}|\mathbf{x}; \mathcal{S}) = \log p(y_i|x_i; \mathcal{D} \cup \mathcal{S}) - \log p(y_i|x_i; \mathcal{S}).$$
$$\tag{5}$$

The fist line in Eq. (5) is obtained by Bayes rule and conditional independence. The second line indicates $G_i = \log p(y_i|x_i; \mathcal{D} \cup \mathcal{S}) - \log p(y_i|x_i; \mathcal{S})$.

From the conclusion in Liu et al. (2019), a vanilla neural network is a special case of a Bayesian neural network with a uniform prior distribution and a Dirac-Delta posterior distribution. Predictive distribution in Eq. (5) could be approximated by vanilla neural network, and the performance gain of vanilla neural network is

$$G_i = \log p(y_i|x_i; \boldsymbol{\theta}_{\mathcal{D} \cup \mathcal{S}}) - \log p(y_i|x_i; \boldsymbol{\theta}_{\mathcal{S}}) = \ell(x_i, y_i; \boldsymbol{\theta}_{\mathcal{S}}) - \ell(x_i, y_i; \boldsymbol{\theta}_{\mathcal{D} \cup \mathcal{S}}), \tag{6}$$

where $\boldsymbol{\theta}_{\mathcal{D} \cup \mathcal{S}}$ and $\boldsymbol{\theta}_{\mathcal{S}}$ denote parameters of model trained on $\mathcal{D} \cup \mathcal{S}$ and $\mathcal{S}$ respectively. Based on the fact that loss in classification task is negative log-probability, $G_i$ can be approximated by loss difference, i.e., ReL. A detailed derivation is shown in Appendix A.. Since $\mathcal{S} \subset \mathcal{D}$ and $|\mathcal{S}| \ll \mathcal{D}$, we approximate $\boldsymbol{\theta}_{\mathcal{D} \cup \mathcal{S}}$ by $\boldsymbol{\theta}_{\mathcal{D}}$ and refer the model trained on $\mathcal{D}$ as holdout model.

Using ReL as the selection criterion, we construct the coreset in a greedy, incremental manner. Initially, we train a holdout model on $\mathcal{D}$ and start with an empty set $\mathcal{S}$. In each step, we initialize and train the model on the current $\mathcal{S}$ to obtain $\boldsymbol{\theta}_{\mathcal{S}}$. Samples with the maximum ReL are selected into $\mathcal{S}$ until the subset reaches the predefined size. We refer to our coreset selection method as CSReL, and detailed coreset selection procedure is shown in Alg. 1.

### 3.3 DISCUSSION ON ReL AND CSReL

We demonstrate in Appendix B that ReL in Eq. (6) is an proportional approximation of the negative sample-weight implicit gradient with mild assumptions. Both ReL and implicit gradients have large

values on representative and informative samples. For ReL, a sample $(x_i, y_i)$ is representative if $\ell(x_i, y_i; \boldsymbol{\theta}_{\mathcal{D}})$ is low and informative if $\ell(x_i, y_i; \boldsymbol{\theta}_{\mathcal{S}})$ is high, indicating new information to $\mathcal{S}$. From the implicit gradient perspective, adding such samples to $\mathcal{S}$ reduces loss on multiple samples after training model on $\mathcal{S} \cup (x_i, y_i)$, resulting in a high absolute implicit gradient value for this sample.

One advantage of our CSReL method is its robustness against non-representative samples, such as ambiguous or noisy ones. In ReL, these samples tend to have high loss values on both $\boldsymbol{\theta}_{\mathcal{D}}$ and $\boldsymbol{\theta}_{\mathcal{S}}$, leading to lower ReL values. Compared to implicit gradient, ReL excludes ambiguous or noisy samples from the indication of holdout model, even if these samples have large loss values on $\boldsymbol{\theta}_{\mathcal{S}}$. Since ReL approximates performance gain, we can conclude that samples with high performance gain are representative and contribute relatively new knowledge, rather than being ambiguous or noisy. Another advantage of ReL is that ReL only requires one forward pass without any backward computation, making it more efficient. In comparison, the implicit gradient necessitates computing the inverse Hessian matrix, which is computationally expensive.

---

**Algorithm 1:** The proposed CSReL

---

**Input:** Dataset $\mathcal{D}$, coreset size $m$, steps $t_{out}$
**Result:** Coreset $\mathcal{C}$
Train holdout model $\boldsymbol{\theta}_{\mathcal{D}} = \arg\min_{\boldsymbol{\theta}} \mathcal{L}(\boldsymbol{\theta})$;
Initialize $\mathcal{S}_0 = \emptyset$; select size of one step
$\quad n = m/t_{out}$;
**for** $k = 1$ **to** $t_{out}$ **do**
$\quad$ Train model on current subset
$\quad\quad \boldsymbol{\theta}_{\mathcal{S}} = \arg\min_{\boldsymbol{\theta}} \hat{\mathcal{L}}(\boldsymbol{\theta})$;
$\quad$ Compute ReL
$\quad\quad G_i = \ell(x_i, y_i; \boldsymbol{\theta}_S) - \ell(x_i, y_i; \boldsymbol{\theta}_{\mathcal{D}})$,
$\quad\quad (x_i, y_i) \in \mathcal{D} \backslash \mathcal{S}_k$;
$\quad$ Select top-$n$ samples $T_k$ by ReL and
$\quad$ update current coreset
$\quad\quad \mathcal{S}_{k+1} = \mathcal{S}_k \cup T_k$;
$\mathcal{C} = \mathcal{S}_k$

---

ReL was first introduced in Mindermann et al. (2022) for training data scheduling. In our work, we show that ReL is well-suited for the coreset selection task and can effectively address the issue of selecting ambiguous or noisy samples in previous coreset selection methods. A detailed discussion comparing CSReL with other related works is provided in Appendix C.

## 4 CORESET SELECTION FOR CONTINUAL LEARNING

### 4.1 APPLYING CORESET SELECTION TO CONTINUAL LEARNING

Continual learning seeks to adapt a CL model to a sequence of tasks with no shared classes between tasks, where data from previous tasks is unavailable during the training of the current task. In this work, we denote the dataset of the $t$-th task as $\mathcal{D}_t$. We focus on the challenging class-incremental setting, where the model has a single classification head, and task identity is not provided during inference. Rehearsal-based methods apply a memory buffer $\mathcal{M}$ to store part of the previous data and replay the stored data during training the CL model to prevent forgetting previous tasks.

We aim to summarize an informative subset as memory for rehearsal-based CL and a natural approach is to select coresets from each task as memory, as proposed in Greedy Coreset (Borsos et al., 2020) and PBCS (Zhou et al., 2022b). We introduce our CSReL continual learning method (CSReL-CL), which selects a coreset from each task using Alg. 1 and equally assign memory size to previous tasks. Detailed algorithm is shown in Appendix E. We illustrate the overview of our CSReL-CL approach in Appendix Q to facilitate a clear understanding of our method.

### 4.2 CONSIDERING PREVIOUS TASKS WHILE UPDATING MEMORY

Only to summarize single task data $\mathcal{D}_t$ for memory may result in interference between tasks, since there may be samples that represent $\mathcal{D}_t$ but interfere other previous tasks. We aim to select samples from $\mathcal{D}_t$ to represent both the current task data and the previous data, so that the optimal subset can represent $\mathcal{D}_t$ while minimizing interference with the previous data. This approach helps to avoid selecting these harmful samples. Since previous data is not available, we use memory data to represent the previous data. We modify performance gain for selecting $i$-th sample as

$$G_i^{\text{Prv}} = \log p\left(\mathbf{y}_{1:t} | \mathbf{x}_{1:t}; \mathcal{S}_t \cup (x_i, y_i)\right) - \log p\left(\mathbf{y}_{1:t} | \mathbf{x}_{1:t}; \mathcal{S}_t\right), \tag{7}$$

where $(\mathbf{x}_{1:t}, \mathbf{y}_{1:t})$ denotes all samples in $\mathcal{D} \cup \mathcal{M}$ and $(x_i, y_i)$ denotes the $i$-th sample in $\mathcal{D}_t$. $\mathcal{S}_t$ is current selected subset from $\mathcal{D}_t$. Following derivation in Section 3, the ReL is

$$G_i^{\text{Prv}} = \ell(x_i, y_i; \boldsymbol{\theta}_{\mathcal{S}_t}) - \ell(x_i, y_i; \boldsymbol{\theta}_{\mathcal{D}_t \cup \mathcal{M}}). \tag{8}$$

The holdout model is trained on both current task data and memory. In practice, since $|\mathcal{M}| \ll |\mathcal{D}_t|$, we resample memory data for training holdout model. We use the abbreviation CSReL-CL-Prv to denote our CL method that considers previous tasks, detailed algorithms are shown in Appendix F.

### 4.3 Selecting Coreset from Streaming Data

Previous summarization methods require the availability of all current data, which may not be practical for large datasets or privacy-related datasets. Reservoir sampling (Vitter, 1985) is an effective method for updating samples from a stream of data without needing the full dataset. Our key idea is to scale up selection probability in reservoir sample so that sample with higher ReL could have higher probability to be selected, while keeping the streaming nature of reservoir sampling.

Since $\mathcal{D}_t$ is not available and the CL model is trained on $\mathcal{D}_t \cup \mathcal{M}$, we approximate $\boldsymbol{\theta}_{\mathcal{D}_t \cup \mathcal{M}}$ by the CL model in Eq. (8) and modify ReL for reservoir sampling as

$$G_i^{\text{RS}} = \ell(x_i, y_i; \boldsymbol{\theta}_{\mathcal{S}_t}) - \ell(x_i, y_i; \boldsymbol{\theta}_{cnt}), \tag{9}$$

where $\boldsymbol{\theta}_{cnt}$ is parameters of the CL model, $\mathcal{S}_t$ is selected subset of current task and $(x_i, y_i)$ denotes $i$-th sample in current batch. ReL in Eq. (9) acts as a scaling factor on the update probability. The selected samples randomly replace existing samples in the memory. We refer to our modified reservoir sampling method as CSReL-RS. The detailed algorithm is provided in Appendix G.

### 4.4 Coreset Selection for Knowledge Distillation

Knowledge distillation (KD) (Hinton et al., 2015) is an effective and widely used method for rehearsal-based CL. In these methods (Buzzega et al., 2020; Yan et al., 2022), the previous samples' logits from the previous models are also save in memory, and a KD loss is applied to regularize the model to produce similar logits for the previous samples.

We claim that coreset selection should take into account both label prediction and KD. When selecting samples in task $t$, we use $(x_i, y_i, o_i)$ to denote the $i$-th sample in $\mathcal{D}_t$, where $o_i$ is the logit of this sample provided by the current CL model. Based on Section 3, we define the performance gain as

$$G_i^{\text{KD}} = \log p\left(\mathbf{o}_{1:t} | \mathbf{x}_{1:t}; \mathcal{S} \cup (x_i, o_i)\right) - \log p\left(\mathbf{o}_{1:t} | \mathbf{x}_{1:t}; \mathcal{S}\right), \tag{10}$$

where $(\mathbf{x}_{1:t}, \mathbf{o}_{1:t})$ denote all sample-logit pairs in $\mathcal{D} \cup \mathcal{M}$. To make the probability of the logit tractable, we assume the probability of the predicted logit follows a Multivariate Gaussian Distribution with an identity covariance matrix. Following Section 3, we can compute ReL for KD as

$$G_i^{\text{KD}} = \ell_{\text{MSE}}(x_i, o_i; \boldsymbol{\theta}_{\mathcal{S}}) - \ell_{\text{MSE}}(x_i, o_i; \boldsymbol{\theta}_{\mathcal{D}_t \cup \mathcal{M}}), \tag{11}$$

where $\ell_{\text{MSE}}$ is mean square error (MSE). To apply objective in Eq. (11) to reservoir sampling, we assume the CL model is trained on $\mathcal{D}_t \cup \mathcal{M}$, as the logits are provided by the CL model and the MSE of this model is 0. Linearly combined ReL (ReL-cmb) with factors $\alpha_s$ and $\beta_s$ is

$$G_i^{\text{cmb}} = \alpha_s G_i^{\text{RS}} + \beta_s G_i^{\text{KD}}. \tag{12}$$

## 5 Experiments

### 5.1 Data Summarization

We evaluate our CSReL selection method on MNIST (Deng, 2012), CIFAR-10 (Krizhevsky et al., 2009) and more challenging CIFAR-100 (Krizhevsky et al., 2009). For MNIST and CIFAR-10, to make a fair comparison, we follow the settings of Borsos et al. (2020). For CIFAR-100, we use ResNet-18 (He et al., 2016) as backbone.

We compare our method with competitive baselines: Greedy Coreset (Borsos et al., 2020), PBCS (Zhou et al., 2022b), we also use uniform sampling as the worst-case. The evaluation metric is the test accuracy of the model trained from scratch on the selected data. Detailed model and training setting are shown in Appendix W.

We plot test accuracy against selected subset size in Figure 1, the results demonstrate that our method performs on par with Greedy Coreset on MNIST dataset and outperforms all baseline methods on CIFAR-10 and CIFAR-100 datasets. Notably, our method performs better with larger coreset sizes. Further comparisons and explanations are shown in Appendix V. Analysis on difficulty of selected samples by our method and Greedy Coreset in Appendix I demonstrate that our method could identify discriminative samples while avoiding selecting ambiguous or noisy samples.

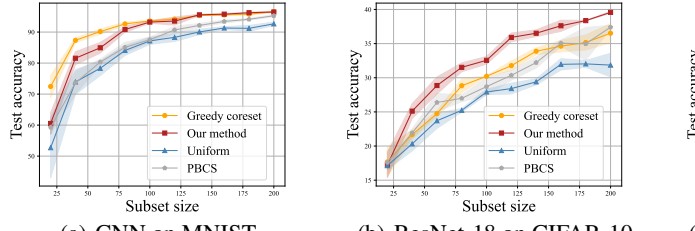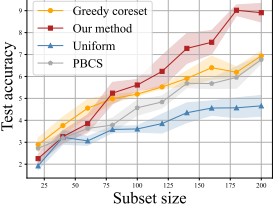

| (a) CNN on MNIST | (b) ResNet-18 on CIFAR-10 | (c) ResNet-18 on CIFAR-100 |

Figure 1: Performance comparisons in data summarization task show that our method performs on par with Greedy Coreset on the MNIST dataset, while outperforming all other baselines on the CIFAR-10 and CIFAR-100 datasets.

Table 1: Final average accuracy, with red and blue indicating the top and second-best values. Our memory construction method performs on par with Greedy Coreset on the MNIST dataset and consistently outperforms other baselines on more complex datasets.

| Methods | Split MNIST | Split CIFAR-10 | Split CIFAR-100 | Perm MNIST |
|---|---|---|---|---|
| Uniform sampling | 93.60±0.66 | 37.05±3.06 | 13.82±1.31 | 78.38±0.82 |
| $k$-means of features | 93.56±1.24 | 35.78±0.56 | 14.31±0.54 | 78.08±0.53 |
| $k$-center of embeddings | 94.03±1.22 | 36.78±4.05 | 14.59±0.32 | 77.93±0.32 |
| Hardest samples | 87.26±2.50 | 27.80±1.20 | 12.19±0.05 | 77.04±0.60 |
| iCaRL's selection | 94.32±0.20 | 35.38±3.12 | 14.43±0.51 | 78.87±0.23 |
| OCS | 84.86±2.69 | 37.12±1.94 | 13.27±0.45 | 75.07±0.91 |
| Greedy Coreset | **95.73±0.19** | 37.68±2.63 | 15.04±0.48 | 79.23±0.37 |
| GCR | 93.22±1.04 | 37.59±1.29 | 13.54±1.06 | 78.73±0.13 |
| PBCS | 94.22±0.61 | 38.37±1.01 | 16.20±0.27 | 76.30±0.81 |
| BCSR | 93.81±0.91 | 38.14±3.64 | 15.11±1.24 | 78.30±0.81 |
| CSReL-CL | **95.68±0.35** | **38.97±2.61** | **17.48±0.21** | **79.59±0.38** |
| CSReL-CL-Prv | 95.55±0.13 | **39.82±0.83** | **18.47±0.17** | **80.02±0.12** |

## 5.2 CONTINUAL LEARNING

We conduct experiments on different CL settings to evaluate our methods, and use the final average accuracy as evaluation metric, which reflects average accuracy across all tasks after training model on the last task. All of our experimental results are obtained from multiple runs with different random seeds. We selecte hyperparameters with a focus on both performance and robustness.

### 5.2.1 CORESET SELECTION FOR CONTINUAL LEARNING

We evaluate CSReL-CL and CSReL-CL-Prv on Split MNIST (Zenke et al., 2017), Perm MNIST (Goodfellow et al., 2013), Split CIFAR-10 and CIFAR-100. To make a fair comparison, we follow the setting of Borsos et al. (2020) and Zhou et al. (2022b). We set 100 memory size for Split MNIST and Perm MNIST and 200 memory size for Split CIFAR-10 and Split CIFAR-100. For Split-CIFAR-100 dataset, we split totally 100 classes into 10 disjoint tasks, and use ResNet-18 (He et al., 2016) as backbone. Detailed experiment settings and hyperparameters are shown in Appendix X.1.

Baselines include: uniform sampling, $k$-center clustering in last layer embedding (Sener & Savarese, 2017) and feature space (Nguyen et al., 2017), iCaRL's selection (Rebuffi et al., 2017), hardest-to-classify samples (Aljundi et al., 2019b), Greedy Coreset (Borsos et al., 2020), GCR (Tiwari et al., 2022), OCS (Yoon et al., 2021), PBCS (Zhou et al., 2022b) and BCSR (Hao et al., 2024).

The average accuracies in Table 1 show that our method performs comparably to Greedy Coreset on the Split MNIST dataset and outperforms all other baselines on the remaining datasets, highlighting the superiority of our approach. Additionally, the experimental results indicate that considering previous tasks can further enhance CL performance. We further demonstrate the effectiveness of our method compared to most recent baselines by conducting additional experiments under the same settings as BCSR, as detailed in Appendix M. The scalability and effectiveness of our method with complex backbone models are further demonstrated in Appendix T.

Table 2: Final average accuracy with red and blue indicating the top and second-best values. Summarizing data with our CSReL-RS consistently improves the performance of existing continual learning methods, particularly in knowledge distillation scenarios.

| Methods | Split CIFAR-100 | | Split Tiny ImageNet | |
|---|---|---|---|---|
| | 200 | 500 | 200 | 500 |
| A-GEM | 9.40±0.05 | 9.42±0.08 | 8.07±0.08 | 8.06±0.04 |
| ER | 14.18±0.45 | 21.08±0.16 | 8.49±0.16 | 9.99±0.29 |
| FDR | 15.32±0.73 | 22.83±0.73 | 8.70±0.19 | 10.54±0.21 |
| CSReL-ER | 15.35±0.73 | 22.65±0.81 | 8.66±0.06 | 10.44±0.17 |
| DER | 21.58±1.72 | 35.20±0.84 | 11.87±0.78 | 17.75±1.14 |
| DER++ | 26.27±2.32 | 36.00±1.92 | 10.96±0.17 | 19.38±1.41 |
| LODE-DER++ | **27.96±0.91** | 39.14±0.74 | 14.46±0.90 | 21.15±0.68 |
| CSReL-DER++ | 27.79±0.60 | **39.80±1.45** | **16.78±0.78** | **21.22±0.92** |
| CSReL-LODE-DER++ | **28.51±0.33** | **41.96±0.78** | **17.01±0.43** | **22.83±0.23** |

### 5.2.2 CORESET SELECTION FOR EXISTING CONTINUAL LEARNING METHODS

To further demonstrate the effectiveness and compatibility of our method with other CL approaches, we replace reservoir sampling in ER (Riemer et al., 2018), DER++ (Buzzega et al., 2020), and LODE-DER++ (Liang & Li, 2024) with our CSReL-RS, resulting in CSReL-ER, CSReL-DER++, and CSReL-LODE-DER++. Since DER++ applies knowledge distillation, we apply ReL-cmb for sample selection for all methods involving DER++. Following settings of the Mammoth framework (Buzzega et al., 2020), we evaluate our methods on Split CIFAR-100 and Split Tiny ImageNet (Wu et al., 2017) with memory sizes of 200 and 500. Both datasets are equally split into 10 tasks with ResNet-18 serving as backbone model. Detailed settings are shown in Appendix X.2.

We compare our methods with other commonly compared rehearsal-based methods including ER, A-GEM (Chaudhry et al., 2018b), FDR (Benjamin et al., 2018), DER (Buzzega et al., 2020), DER++ and LODE-DER++. The results are presented in Table 2, with baseline method results on Split Tiny ImageNet taken from Buzzega et al. (2020) and Liang & Li (2024).

The results in Table 2 show that summarizing the coreset using our CSReL-RS effectively enhances existing continual learning methods, particularly when applying knowledge distillation. Combining CSReL-RS with LODE-DER++ consistently outperforms other methods across all datasets. These findings highlight both the effectiveness and compatibility of our approach. Additionally, we evaluate the scalability of our method on a 500-class subset of ImageNet-1K in Appendix N.

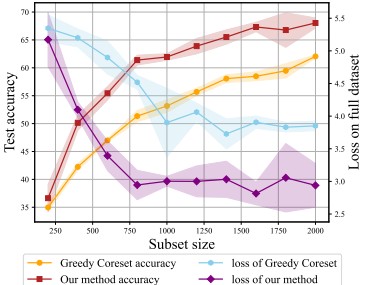

### 5.3 ABLATION STUDY

**Analysis on representing full dataset.** Both Greedy Coreset (Borsos et al., 2020) and our method minimize loss on the full dataset of model trained on coreset. A lower loss indicates that coreset better represents the full dataset. To evaluate this, we train models on the selected subsets and compute the average loss on the full dataset as a metric to assess how well the selected data represents

Figure 2: Average loss on the full dataset and test accuracy of models trained on subsets show that coreset selected by our method could better represent full dataset and achieve higher accuracy.

the full dataset. We conduct experiments on CIFAR-10 by selecting 2,000 samples from a pool of 10,000 with our method and Greedy Coreset. Models are trained on subsets with different sizes during the selection process, and we evaluate the trained models based on both the average loss on the full dataset and test set accuracy.

We plot test accuracy and average loss on full dataset against subset size in Figure 2, The results indicate that models trained on subsets selected by our method achieve a greater reduction in average full dataset loss, especially when the subset size is small. This suggests that our method selects samples with higher performance gains and thus better represent the full dataset. The corresponding test accuracy is consistent with the loss curve, demonstrating the effectiveness of our method.

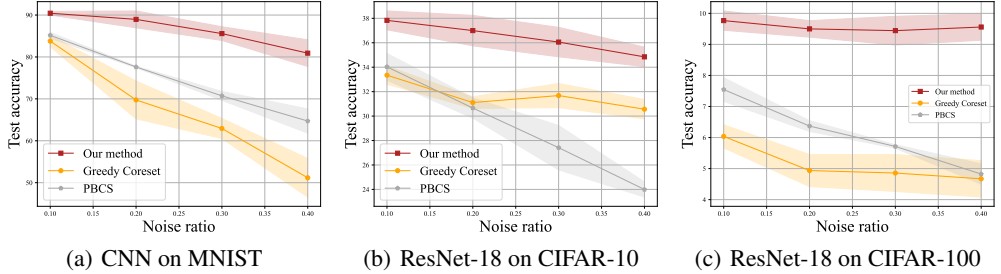

| (a) CNN on MNIST | (b) ResNet-18 on CIFAR-10 | (c) ResNet-18 on CIFAR-100 |

Figure 4: Test accuracy under different noise ratios shows that the performance of our method drops only slightly as the noise ratio increases and outperforms other baselines in noisy condition.

**Data summarization under label noise.** To demonstrate the robustness of our method under noisy condition, we test data summarization performance under label noise case on MNIST, CIFAR-10 and CIFAR-100 dataset, with the same experiment setting and evaluation method as experiments in Section 5.1. Specifically, we randomly corrupt samples at different ratios and then use this corrupted dataset as the selection pool to select the coreset of 200 samples. We compare our method with

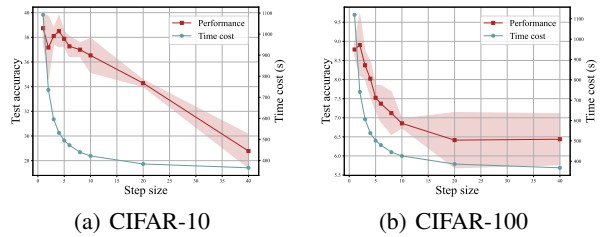

| (a) CIFAR-10 | (b) CIFAR-100 |

Figure 3: Performance and time cost w.r.t. step size. Selecting more samples within one step will degrade performance while reducing time cost, indicating that a trade-off should be made between performance and efficiency.

Greedy Coreset (Borsos et al., 2020) and PBCS (Zhou et al., 2022b). No clean holdout set is provided, namely, holdout model in our method is trained on noisy dataset, outer loss of Greedy Coreset and PBCS is computed on noisy dataset.

As shown in Figure 4, the performance of our method drops slightly as the noise ratio increases across all datasets, while the performance of other methods drops significantly. We also count number of selected noisy samples in Appendix K, showing that our method selects significantly fewer noisy samples. These results demonstrate the robustness of our method to data noise. Given that low noise ratios are common in practical scenarios, such as web data, our approach effectively avoids selecting noisy data for the coreset.

**Selection steps in data summarization.** In our multi-step selection algorithm, selecting more samples in one step and using fewer steps can reduce computation costs. However, selecting more samples in one step may result in redundant samples, as samples with similar ReL may contain similar knowledge. Therefore, a trade-off should be made between efficiency and performance. We select coresets of 200 samples with different step size on CIFAR-10 and CIFAR-100 and train models on these coresets. Test accuracy of models and time cost corresponding to step size are shown in Figure 3.

We observe that performance decreases as the step size increases, while the time cost decreases. These results verify our claim of a trade-off between time cost and performance. For CIFAR-10 dataset, performance drops slightly when the step size is smaller than 10. However, in the more complex CIFAR-100 dataset, performance drops significantly as the step size increases.

**Effectiveness of considering knowledge distillation.** To verify the effectiveness of considering knowledge distillation proposed in Section 4.4, we set different MSE factor $\beta_s$ while fixing all other hyper parameters, $\alpha_s$ is set to 1.0 in all the experiments. We conduct experiments on Split CIFAR-100 with CSReL-RS, and use ReL-cmb in Eq. (12) for selection. Figure 5 shows the average accuracy and forgetting w.r.t. different $\beta_s$ values.

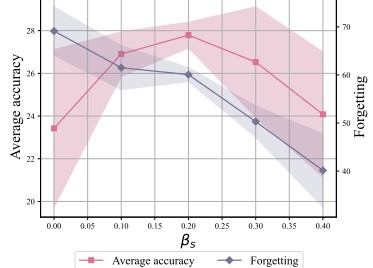

Figure 5: Avg. accuracy and forgetting under different $\beta_s$. Increasing $\beta_s$ decreases forgetting and the best accuracy is achieved when $\beta_s$ is set to 0.2.

As $\beta_s$ increases, forgetting continually decreases, indicating that more emphasis is placed on selecting samples that encourage the current model to mimic the output of the previous model, thereby increasing regularization strength. The best performance is achieved when $\beta_s = 0.2$, suggesting that excessive regularization can hinder the learning of new tasks, highlighting the need for balance. Therefore, using ReL-cmb proves effective for coreset selection.

**Holdout model training epochs.** In our work, holdout model serves as an indicator for which sample is worthy of learning and which sample is not learned yet, we test the influence of holdout model training epochs with respect to the quality of selected samples on CIFAR-10. Specifically, we train holdout models with different epochs, then select coresets with our method and train models on selected coresets. We evaluate models with test accuracy. Coreset size is set to 200, backbone and other hyperparameters are the same as Section 5.1.

From result in Table 3, as holdout model training epochs increases, quality of selected coreset increases and remains stable after holdout training epochs reaches 80. Indicating that quality of holdout model affects quality of selected data, and holdout model should be well trained for coreset selection. We also evaluate the impact of holdout model training epochs on the continual learning task using the Split CIFAR-100 dataset, as detailed in Appendix O. Our results show that continual learning performance remains stable across a wide range of holdout model training epochs.

Table 3: Performance with respect to different holdout training epochs in the data summarization task: Performance increases as training epochs increase and then remains stable.

| Train epochs | Test accuracy |
|---|---|
| 20 | 32.98±0.75 |
| 40 | 35.13±0.75 |
| 60 | 35.12±0.63 |
| 80 | 37.57±0.78 |
| 100 | 38.74±1.08 |
| 120 | 37.89±0.27 |

**Time cost of coreset selection.** To demonstrate the efficiency of our CSReL coreset selection method, we provide the time costs for coreset selection on different datasets in Section 5.1, as shown in Table 4. The time cost of our method scales moderately across different datasets and backbones, and training the holdout model remains manageable with larger datasets and backbones. This demonstrates that our method is well-suited for larger datasets and more complex backbones.

Table 4: Time cost of selecting coreset with CSReL. The time cost scales mildly on different datasets.

| Dataset | Holdout train | Selection |
|---|---|---|
| MNIST | 159.84s | 102.70s |
| CFIAR-10 | 349.74s | 737.68s |
| CIFAR-100 | 350.87s | 735.85s |

Both our method and Greedy Coreset (Borsos et al., 2020) use matching pursuit for coreset selection. However, Greedy Coreset computes the *Neural Tangent Kernel (NTK)* (Jacot et al., 2018), which takes over 4000 seconds. In comparison, our CSReL coreset selection method is significantly faster, demonstrating its efficiency. All our experiments are conducted on NVIDIA RTX3090. Details of backbone parameters and time cost is shown in Appendix P.

**Further ablation studies.** In greedy incremental selection, selected samples are not removed, meaning that samples chosen early may have a lower performance gain as more samples are added to the coreset. As a result, the selection order can influence the performance of the coreset. We demonstrate the robustness of our method with respect to selection order in Appendix R. Additionally, difficult and noisy samples cannot be distinguished solely using the ReL metric. However, in greedy incremental selection, we show that our method can effectively differentiate between these sample types in Appendix S. Furthermore, we present a feature map of the selected samples in Appendix U, illustrating that our method consistently selects more representative samples.

## 6 CONCLUSION

In this work, we address the problem of selecting ambiguous or noisy samples in previous bilevel coreset selection methods by using reducible loss as an approximation for performance gain. This approach enables the identification of representative and informative samples while excluding noisy or ambiguous ones, leading to improved performance. Additionally, we propose an efficient coreset selection method designed to address the unique challenges of continual learning, such as task interference, streaming scenarios, and knowledge distillation. Extensive experiments validate the effectiveness of our approach in both data summarization and continual learning tasks. In future work, we plan to explore coreset selection for fine-tuning large pretrained models, allowing them to acquire new knowledge while maintaining generalizability.

ACKNOWLEDGEMENTS

This paper was partially supported by Australian Research Council (ARC) Discovery Early Career Researcher Award (DECRA) project DE230101591 to D. Gong.

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

# Appendix

## Table of Contents

## A   Approximating Predictive Distribution with Reducible Loss

Predictive distribution in Eq. (5) requires parameter distribution of model trained on $\mathcal{D} \cup \mathcal{S}$ and $\mathcal{S}$

$$p\left(y_i | x_i; \mathcal{D} \cup \mathcal{S}\right) = \int p\left(y_i | x_i; \boldsymbol{\theta}\right) p\left(\boldsymbol{\theta} | \mathcal{D} \cup \mathcal{S}\right) d\boldsymbol{\theta},$$

$$p\left(y_i | x_i; \mathcal{S}\right) = \int p\left(y_i | x_i; \boldsymbol{\theta}\right) p\left(\boldsymbol{\theta} | \mathcal{S}\right) d\boldsymbol{\theta}.$$

From the conclusion in (Liu et al., 2019), a vanilla neural network is a special case of a Bayesian neural network with a uniform prior distribution and a Dirac-Delta posterior distribution. Therefore, performance gain of vanilla neural network is

$$G_i = \log p\left(y_i | x_i; \boldsymbol{\theta}_{\mathcal{D} \cup \mathcal{S}}\right) - \log p\left(y_i | x_i; \boldsymbol{\theta}_{\mathcal{S}}\right), \tag{13}$$

where $\boldsymbol{\theta}_{\mathcal{D} \cup \mathcal{S}}$ and $\boldsymbol{\theta}_{\mathcal{S}}$ denote parameters of model trained on $\mathcal{D} \cup \mathcal{S}$ and $\mathcal{S}$ respectively. For classification task and cross-entropy loss, we use loss to replace the negative log-probability, the performance gain is

$$G_i = \ell(x_i, y_i; \boldsymbol{\theta}_{\mathcal{S}}) - \ell(x_i, y_i; \boldsymbol{\theta}_{\mathcal{D} \cup \mathcal{S}}). \tag{14}$$

## B   Relation between ReL and Implicit Gradient

We show that ReL in Eq. (14) is a proportional approximation to negative implicit gradient in (Borsos et al., 2020) under greedy selection framework with binary sample weight. Given model trained on $\mathcal{S}$, parameter $\boldsymbol{\theta}_{\mathcal{D} \cup \mathcal{S}}$ is approximated by updating $\boldsymbol{\theta}_{\mathcal{S}}$ with one Newton step, namely

$$\boldsymbol{\theta}_{\mathcal{D}} \approx \boldsymbol{\theta}_{\mathcal{S}} - \mathbf{H}_{full}^{-1} \nabla \mathcal{L}(\boldsymbol{\theta}_{\mathcal{S}}), \tag{15}$$

where $\mathbf{H}_{full}$ denotes Hessian matrix of loss $\sum_{(x_i, y_i) \in \mathcal{D}} \ell(x_i, y_i; \boldsymbol{\theta}_{\mathcal{S}})$ with respect to $\boldsymbol{\theta}_{\mathcal{S}}$. Since $\mathcal{S} \subset \mathcal{D}$ and samples are i.i.d., we could assume that the Hessian matrix $\mathbf{H}$ provides a statistical approximation of the full Hessian $\mathbf{H}_{full}$ in the neighbor space of $\boldsymbol{\theta}_{\mathcal{S}}$, the following relationship holds

$$\frac{1}{|\mathcal{D}|} \mathbf{H}_{full} \approx \frac{1}{|\mathcal{S}|} \mathbf{H}.$$

Approximating $G_i$ in Eq. (14) with first-order Taylor expansion and substituting parameter difference in Eq. (15) results in

$$G_i \approx \nabla \ell(x_i, y_i; \boldsymbol{\theta}_{\mathcal{S}})^T (\boldsymbol{\theta}_{\mathcal{S}} - \boldsymbol{\theta}_{\mathcal{D}}) \approx \frac{|\mathcal{S}|}{|\mathcal{D}|} \nabla \ell(x_i, y_i; \boldsymbol{\theta}_{\mathcal{S}})^T \mathbf{H}^{-1} \nabla \mathcal{L}(\boldsymbol{\theta}_{\mathcal{S}}). \tag{16}$$

Since CSR is computed over all candidate samples and the ratio $|\mathcal{S}|/|\mathcal{D}|$ does not affect the ranking of candidate samples, Eq. (16) indicates ReL is a proportional approximation to negative implicit gradient in Greedy Coreset (Borsos et al., 2020).

For samples which are representative and not well represented by $\mathcal{S}$, both implicit gradient and ReL have high scores. Suppose $(x_i, y_i)$ is such kind of sample, for ReL, representativeness indicates $\ell(x_i, y_i; \boldsymbol{\theta}_{\mathcal{D}})$ is low, and not represented by $\mathcal{S}$ means $\ell(x_i, y_i; \boldsymbol{\theta}_{\mathcal{S}})$ is high, thus ReL is high. From the aspect of implicit gradient, adding these samples into $\mathcal{S}$ will reduce loss on multiple samples after training model on $\mathcal{S} \cup (x_i, y_i)$ since this sample contains knowledge similar to other samples that is not yet present in $\mathcal{S}$. According to the definition of implicit gradient $d\mathcal{L}(\boldsymbol{\theta}_{\mathcal{S}})/dw_i$, a reduction in loss across multiple samples indicates a high absolute value of the implicit gradient for that sample.

## C  DISCUSSION WITH OTHER REDUCIBLE LOSS RELATED WORKS

Reducible loss was firstly introduced in Mindermann et al. (2022) for selecting samples that are both learnable and worth learning from each batch of data. Sujit et al. (2023) further applied ReL for selecting samples to replay in reinforcement learning, where unselected samples may be chosen in later epochs or episodes. However, in the coreset selection task, unselected samples are discarded and not reused after the selection process. As a result, the methods proposed in previous works cannot be directly applied to coreset selection task.

Online training data scheduling can also select a subset from each incoming batch for training (Evans et al., 2023), with the goal of maximizing the performance of the model trained on the selected data. The target model is trained with only a single step on this subset, which may prevent it from fully learning the knowledge contained in these samples. When the training dataset is large enough and the pruning rate is not too high, the model can gradually acquire sufficient knowledge, allowing this method to perform well. However, in the coreset selection task, we aim to select a small subset from the entire dataset, meaning the pruning rate is much higher, and the full dataset may not be as large. This can result in lower performance in the coreset selection task.

Our work adopts the same matching pursuit selection framework as Greedy Coreset (Borsos et al., 2020). To address the issue of unselected data being excluded after selection, we train the model on the selected subset until convergence, ensuring that the model fully captures the knowledge contained in the selected data. The incremental selection method ensures that each newly selected sample adds new knowledge to the existing subset.

Our work proves that ReL could act as an indicator for sample selecting in model convergence condition. Eq. (16) shows that ReL is a proportional approximation of the negative implicit gradient, while Eq. (3) illustrates that the implicit gradient computes how changes in sample weights affect the outer loss via the chain rule. The optimal parameter change $d\boldsymbol{\theta}_S/dw_i$ estimates how the model parameters will change when sample weights are modified, assuming the model is trained to convergence. Thus, in the case of a converged model, ReL can serve as a reliable indicator.

Building on the coreset selection method from Greedy Coreset (Borsos et al., 2020), we address the issue of selecting ambiguous or noisy samples using the implicit gradient and directly set the performance gain in Eq. (4) as our selection objective. ReL serves as an approximation of this performance gain. Both ReL and the implicit gradient can identify representative and informative samples. However, compared to the implicit gradient, ReL is more effective at avoiding the selection of ambiguous and noisy samples, resulting in improved performance over Greedy Coreset (Borsos et al., 2020). Therefore, our method could effectively select coreset of representative and informative samples. A related research area involves data compression (Wang et al., 2022b), which seeks to store more information with limited storage capacity by compressing data.

## D  DERIVATION OF OUTER LOSS GRADIENT WITH RESPECT TO SAMPLE WEIGHT

In this section, we derive Eq. (2) from the bilevel optimization formulation presented in Eq ( 1). We follow a logical sequence of steps to arrive at the gradient expression in the desired form. The gradient has been investigated in previous works (Lorraine et al., 2020; Franceschi et al., 2017; Zhang et al., 2024). For completeness, we provide a simplified version here.

**Step 1: First-Order Optimality Condition for $\boldsymbol{\theta}^*(\mathbf{w})$**  We begin by noting that $\boldsymbol{\theta}^*(\mathbf{w})$ is the minimizer of the weighted loss function $\hat{\mathcal{L}}(\boldsymbol{\theta})$, which is defined as:

$$\hat{\mathcal{L}}(\boldsymbol{\theta}) = \sum_{i=1}^{|\mathcal{D}|} w_i \ell(x_i, y_i; \boldsymbol{\theta}),$$

where $\boldsymbol{\theta}$ are the parameters of the model. At the optimal solution $\boldsymbol{\theta}^*(\mathbf{w})$, the gradient of the weighted loss with respect to $\boldsymbol{\theta}$ must vanish:

$$\frac{\partial \hat{\mathcal{L}}(\boldsymbol{\theta})}{\partial \boldsymbol{\theta}}\bigg|_{\boldsymbol{\theta}=\boldsymbol{\theta}^*(\mathbf{w})} = 0.$$

This condition represents the first-order optimality condition for the minimizer of $\hat{\mathcal{L}}(\boldsymbol{\theta})$, implying that at $\boldsymbol{\theta}^*(\mathbf{w})$, the gradient of the weighted loss function with respect to $\boldsymbol{\theta}$ is zero.

**Step 2: Implicit Function Theorem Application** Next, we apply the implicit function theorem to relate the optimal model parameters $\boldsymbol{\theta}^*(\mathbf{w})$ to the weight vector $\mathbf{w}$. Since $\boldsymbol{\theta}^*(\mathbf{w})$ is implicitly defined by the equation

$$\frac{\partial \hat{\mathcal{L}}(\boldsymbol{\theta})}{\partial \boldsymbol{\theta}}\bigg|_{\boldsymbol{\theta}=\boldsymbol{\theta}^*(\mathbf{w})} = 0,$$

we can compute the derivative of $\boldsymbol{\theta}^*(\mathbf{w})$ with respect to $w_i$. Differentiating both sides of the equation with respect to $w_i$ gives:

$$\frac{\partial}{\partial w_i}\left(\frac{\partial \hat{\mathcal{L}}(\boldsymbol{\theta})}{\partial \boldsymbol{\theta}}\bigg|_{\boldsymbol{\theta}=\boldsymbol{\theta}^*(\mathbf{w})}\right) = 0.$$

By the chain rule, this becomes:

$$\frac{\partial^2 \hat{\mathcal{L}}(\boldsymbol{\theta})}{\partial \boldsymbol{\theta}^2}\bigg|_{\boldsymbol{\theta}=\boldsymbol{\theta}^*(\mathbf{w})}\frac{\partial \boldsymbol{\theta}^*(\mathbf{w})}{\partial w_i} + \frac{\partial^2 \hat{\mathcal{L}}(\boldsymbol{\theta})}{\partial \boldsymbol{\theta}\partial w_i}\bigg|_{\boldsymbol{\theta}=\boldsymbol{\theta}^*(\mathbf{w})} = 0.$$

Here, the Hessian matrix $\mathbf{H}$ of $\hat{\mathcal{L}}(\boldsymbol{\theta})$ is the second derivative of $\hat{\mathcal{L}}(\boldsymbol{\theta})$ with respect to $\boldsymbol{\theta}$, evaluated at $\boldsymbol{\theta}^*(\mathbf{w})$. We can thus rewrite this as:

$$\mathbf{H}\frac{\partial \boldsymbol{\theta}^*(\mathbf{w})}{\partial w_i} = -\frac{\partial \hat{\mathcal{L}}(\boldsymbol{\theta})}{\partial w_i}\bigg|_{\boldsymbol{\theta}=\boldsymbol{\theta}^*(\mathbf{w})}.$$

**Step 3: Gradient of $\mathcal{L}(\boldsymbol{\theta}^*(\mathbf{w}))$ with Respect to $w_i$** Now we seek the gradient of the outer loss function $\mathcal{L}(\boldsymbol{\theta}^*(\mathbf{w}))$ with respect to the weight $w_i$. The outer loss function is:

$$\mathcal{L}(\boldsymbol{\theta}^*(\mathbf{w})) = \sum_{i=1}^{|\mathcal{D}|} \ell(x_i, y_i; \boldsymbol{\theta}^*(\mathbf{w})).$$

By the chain rule, the gradient of $\mathcal{L}(\boldsymbol{\theta}^*(\mathbf{w}))$ with respect to $w_i$ is:

$$\nabla_{w_i}\mathcal{L}(\boldsymbol{\theta}^*(\mathbf{w})) = \nabla_{\boldsymbol{\theta}}\mathcal{L}(\boldsymbol{\theta}^*(\mathbf{w}))^T\frac{\partial \boldsymbol{\theta}^*(\mathbf{w})}{\partial w_i}.$$

Substitute the expression for $\frac{\partial \boldsymbol{\theta}^*(\mathbf{w})}{\partial w_i}$ from the implicit function theorem:

$$\nabla_{w_i}\mathcal{L}(\boldsymbol{\theta}^*(\mathbf{w})) = -\nabla_{\boldsymbol{\theta}}\mathcal{L}(\boldsymbol{\theta}^*(\mathbf{w}))^T\mathbf{H}^{-1}\nabla_{\boldsymbol{\theta}}\ell(x_i, y_i; \boldsymbol{\theta}^*(\mathbf{w})).$$

This gives the desired expression for the gradient of the loss function with respect to $w_i$, as provided in Eq. (2).

# E  ALGORITHMS FOR CSReL CONTINUAL LEARNING

In this work, we apply our coreset selection method to rehearsal-based CL. Training objective at task $t$ is

$$\mathcal{L}_{cnt}(\mathcal{D}_t \cup \mathcal{M}; \boldsymbol{\theta}_{cnt}) = \frac{1}{|\mathcal{B}|}\sum \ell(x_i, y_i; \boldsymbol{\theta}_{cnt}) + \alpha\frac{1}{|\mathcal{B}_m|}\sum \ell(x_m, y_m; \boldsymbol{\theta}_{cnt}), \qquad (17)$$

where $(x_i, y_i)$ and $(x_m, y_m)$ are samples from the current task dataset and memory $\mathcal{M}$ respectively. The memory $\mathcal{M}$ stores coresets from all previous tasks, defined as $\mathcal{M} = \cup_{i=1}^{t}\mathcal{C}_i$. $\alpha$ is hyperparameter for balancing regularization force from memory, $\mathcal{B}$ and $\mathcal{B}_m$ are batches from current task and memory respectively.

After training task $t$, we select coreset from $\mathcal{D}_t$ with Alg.1, and store the holdout model loss of each selected sample. To shrink memory of previous data, we re-select memory data to shrink memory of previous tasks, using memory data of previous tasks as selection pool. Since the holdout model loss for previous memory samples are already stored in the memory, there is no need to retrain the holdout model during re-selection. We refer our CL method as CSReL Continual Learning (CSReL-CL) and detailed algorithm is shown in Alg.2.

---

**Algorithm 2:** CSReL Continual Learning

---

**Input:** Dataset sequence $\mathcal{D}_{1:T}$, memory size $K$
Initialize memory $\mathcal{M}_0 = \emptyset$;
**for** *i in range(1,$T + 1$)* **do**
    Train continual model with replay $\boldsymbol{\theta}_{cnt}^* = \arg\min_{\theta} \mathcal{L}_{cnt}(\mathcal{D}_t \cup \mathcal{M}; \boldsymbol{\theta})$;
    `// Update memory`
    Compute size for each task $k_i = K/i, j \in [1:i]$;
    **for** *j in range(i)* **do**
        Reselect samples for previous tasks from $\mathcal{C}_j$ by Alg.1, coreset size is $k_i$;
    Select samples for current task from $\mathcal{D}_i$ by Alg.1, $|\mathcal{C}_t| = k_i$;
    Form new memory $\mathcal{M} = \cup_{j=1}^i \mathcal{C}_j$;

---

## F  ALGORITHMS FOR CSReL-CL CONSIDERING PREVIOUS TASKS

---

**Algorithm 3:** Coreset selection considering previous tasks

---

**Input:** Dataset $\mathcal{D}$, select size $m$, selection steps $t_{out}$, memory $\mathcal{M}$
**Result:** Coreset $\mathcal{C}$
Define holdout loss function: $\mathcal{L}_{hld}(\boldsymbol{\theta}) = 1/(|\mathcal{D}| + |\mathcal{M}|) \sum_{i \in \mathcal{D} \cup \mathcal{M}} \ell(x_i, y_i; \boldsymbol{\theta})$;
Initialize $\mathcal{S}_0 = \emptyset$;
Train holdout model $\boldsymbol{\theta}_{\mathcal{D}} = \arg\min_{\boldsymbol{\theta}} \mathcal{L}_{hld}(\boldsymbol{\theta})$;
Select size of one step $n = m/t_{out}$;
`// Outer loop`
**for** *k in range($t_{out}$)* **do**
    $\boldsymbol{\theta}_{\mathcal{S}} = \arg\min_{\theta} \hat{\mathcal{L}}(\boldsymbol{\theta})$;
    Compute ReL $G_i^{\mathrm{Prv}} = \ell(x_i, y_i; \boldsymbol{\theta}_{\mathcal{S}}) - \ell(x_i, y_i; \boldsymbol{\theta}_{\mathcal{D}}), \quad (x_i, y_i) \in \mathcal{D} \backslash \mathcal{S}_k$;
    Select top-$n$ samples $T_k$ by ReL;
    Update current coreset $\mathcal{S}_{k+1} = \mathcal{S}_k \cup T_k$;
$\mathcal{C} = \mathcal{S}_k$

---

**Algorithm 4:** CSReL-CL-Prv

---

**Input:** Dataset sequence $\mathcal{D}_{1:T}$, memory size $K$
Initialize memory $M^0 = \emptyset$;
**for** *i in range(1,$T + 1$)* **do**
    `// Train continual model with replay`
    $\boldsymbol{\theta}_{cnt}^* = \arg\min_{\theta} \mathcal{L}_{cnt}(\mathcal{D}_t \cup \mathcal{M}; \theta)$;
    `// Update memory`
    Compute size for each task $k_i = K/i, j \in [1:i]$;
    **for** *j in range(i)* **do**
        Reselect samples for task $j$ from $\mathcal{C}_j$ by Alg.3, coreset size is $k_i$;
    Select samples for current task from $\mathcal{D}_i$ with $\mathcal{M}$ by Alg.3, $|\mathcal{C}_t| = k_i$;
    Form new memory $\mathcal{M} = \cup_{j=1}^i \mathcal{C}_j$;

---

## G  ALGORITHMS FOR CSReL RESERVOIR SAMPLING

ReL for selecting sample in data stream is

$$G_i^{\mathrm{RS}} = \ell(x_i, y_i; \boldsymbol{\theta}_{\mathcal{S}_t}) - \ell(x_i, y_i; \boldsymbol{\theta}_{cnt}), \tag{18}$$

where $\boldsymbol{\theta}_{cnt}$ is parameters of continual learning model, $\mathcal{S}_t$ is selected subset of current task and $(x_i, y_i)$ denotes $i$-th sample in $\mathcal{B}_t$.

To apply our selection method to reservoir sampling, we maintain an memory model which is trained on $\mathcal{S}_t$, with parameters $\boldsymbol{\theta}_{\mathcal{S}_t}$. ReL computed in Eq. (18) is applied to modify update probability so that

samples with higher ReL will have higher probability to be selected to memory. Specifically, for each batch, we firstly compute ReL for each sample with Eq. (18), then we normalize ReL by Softmax function as probability scaling factor. The final update probability is original update probability multiplied by scaling factor. Once one sample is selected, this sample randomly replace one existing sample in current memory. To update information contained in $\boldsymbol{\theta}_{\mathcal{S}_t}$, we retrain additional model on selected data of current task when number of newly updated samples reaches threshold $N_{upd}$, and we reinitialize $\boldsymbol{\theta}_{\mathcal{S}_t}$ when a new task comes. We refer to our modified reservoir sampling method as CSReL-RS.

CSReL-RS update process is shown in Alg. 5, where $\mathcal{B}$ is current training batch, $n$ is number of seen samples in this stream, and $K$ is memory size. CL process applying our CSReL-RS is shown in Alg.6.

---

**Algorithm 5:** CSReL-RS

**Input:** Current batch $\mathcal{B} = \{(x_i, y_i)\}$, continual model $\boldsymbol{\theta}_{cnt}$, memory model $\boldsymbol{\theta}_{\mathcal{S}_t}$, number of seen samples $n$, memory size $K$, retrain threshold $N_{upd}$, number of updated samples $n_{upd}$

Compute ReL: $G_i = \ell(x_i, y_i; \boldsymbol{\theta}_{\mathcal{S}_t}) - \ell(x_i, y_i; \boldsymbol{\theta}_{cnt})$;

Compute probability scaling factor: $s_i = \frac{\exp(G_i^{\text{RS}})}{\sum \exp(G_i^{\text{RS}})} \cdot |\mathcal{B}|$;

**for** $i$ *in range* $|\mathcal{B}|$ **do**
    **if** $n < |K|$ **then**
        // Add sample to memory
        $\mathcal{M} = \mathcal{M} \cup (x_i, y_i)$;
        $n_{upd} += 1$;
    **else**
        $r = \text{randint}(0, n+1)$;
        // Scale probability of update by $s_i$
        **if** $r < (K \cdot s_i)$ **then**
            // Replace existing sample
            $pos = \text{randint}(0, |\mathcal{M}|)$;
            $\mathcal{M}[pos] = (x_i, y_i)$;
            $n_{upd} += 1$;
    $n += 1$;
**if** $n_{upd} \geq N_{upd}$ **then**
    Initialize $\boldsymbol{\theta}_{\mathcal{S}_t}$;
    Train $\boldsymbol{\theta}_{\mathcal{S}_t}$ on memory data of current task;
    $n_{upd} = 0$;

---

**Algorithm 6:** Continual learning with CSReL-RS

**Input:** Dataset sequence $\mathcal{D}_{1:T}$, Memory size $K$, loss function $\mathcal{L}_{cnt}$

Initialize memory $M_0 = \emptyset$;
Initialize continual model $\boldsymbol{\theta}_{cnt}$;
**for** $t$ *in range(1, T+1)* **do**
    **for** $\mathcal{B}$ *in* $\mathcal{D}_i$ **do**
        Sample memory batch $\mathcal{B}_{\mathcal{M}}$;
        Update continual model $\boldsymbol{\theta}_{cnt} = \boldsymbol{\theta}_{cnt} - \eta\nabla\mathcal{L}_{cnt}(\mathcal{B}_t \cup \mathcal{B}_{\mathcal{M}}; \boldsymbol{\theta}_{cnt})$;
        // Here we reuse loss on $\mathcal{B}$ before update $\theta_{cnt}$
        Update memory $\mathcal{M}$ by Alg.5;

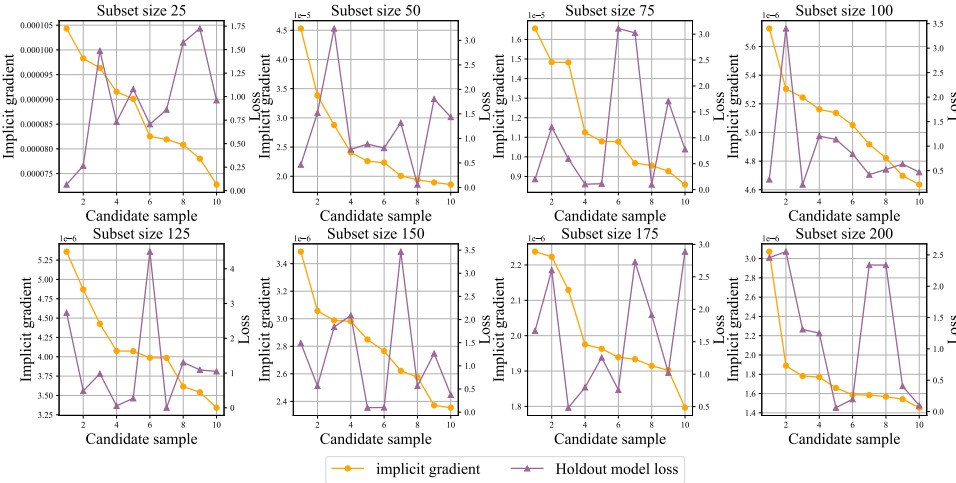

Figure 6: Implicit Gradient and holdout model loss of top-10 candidates in Greedy Coreset selection on CIFAR-10 dataset.

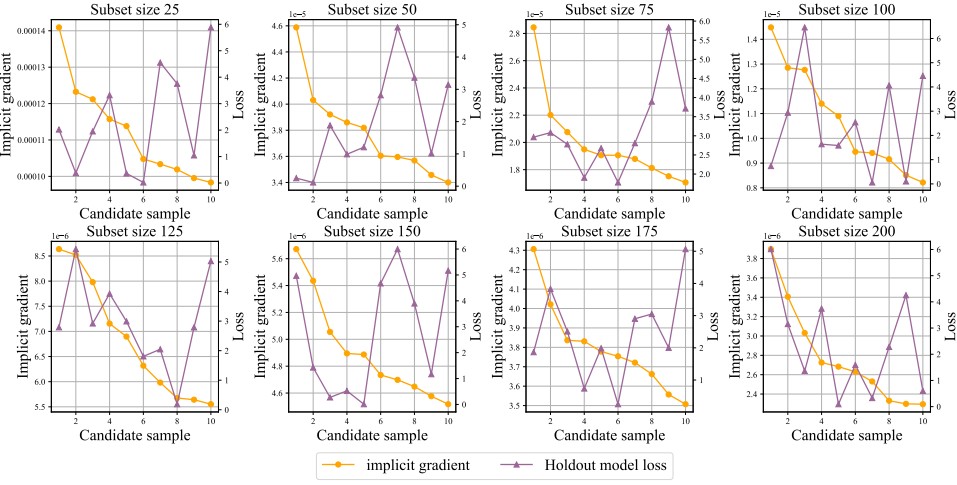

Figure 7: Implicit Gradient and holdout model loss of top-10 candidates in Greedy Coreset selection on CIFAR-100 dataset.

## H   Loss of High Implicit Gradient Candidates

To verify the claim that samples with high implicit gradients may be noisy or ambiguous, we plot the implicit gradient and holdout model loss of the top 10 implicit gradient candidates during the selection process of Greedy Coreset. We use the original implementation of Greedy Coreset for selection, with the holdout model trained on the full selection pool. Our experiments are conducted on the CIFAR-10 and CIFAR-100 datasets, and we plot the candidates after every 25 coreset samples are selected. The results for CIFAR-10 and CIFAR-100 are shown in Figures 6 and 7 respectively.

The x-axis represents the rank of candidate samples, the left y-axis denotes the implicit gradient value, and the right y-axis shows the loss value. From Figure 6, we observe that some samples have high holdout model loss values. After selecting 125 and 200 samples, the top-ranked candidate samples exhibit holdout model losses greater than 2.0, indicating that these samples are almost misclassified by the holdout model. A similar pattern is observed in the CIFAR-100 experiment after selecting 150 and 200 samples as shown in Figure 7.

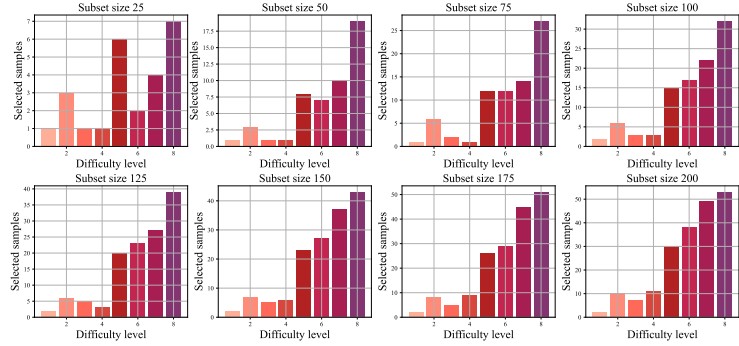

(a) Number of selected sample by our method from each difficulty level during selection on MNIST

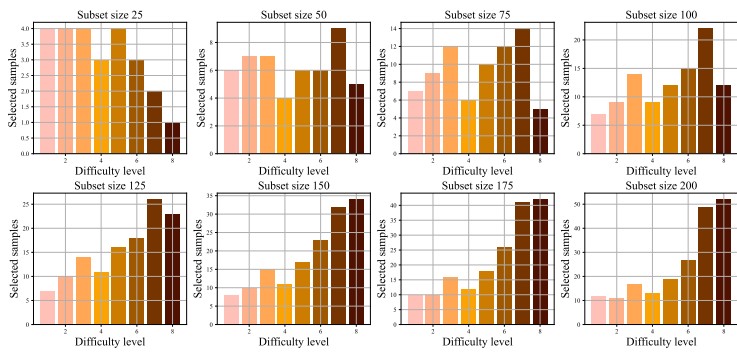

(b) Number of selected sample by Greedy Coreset from each difficulty level during selection on MNIST

Figure 8: Difficulty analysis on MNIST during selection.

Since samples with high holdout model loss are often ambiguous or noisy, these results suggest that such samples may exhibit high implicit gradient values and could be selected for the coreset.

# I  DIFFICULTY OF SELECTED SAMPLES

We analyze difficulty of selected samples. To define sample difficulty, we firstly train a model on full dataset to convergence, then we compute loss on each sample in full dataset, we use this loss as metric of difficulty. This difficulty means how hard-to-learn of one sample. Based on difficulty, we equally split all samples into 8 groups with the incremental of difficulty, indicating 8 levels of difficulty. For each difficulty level, we count how many samples in this level are selected, aiming to analyze preference of one selection method. We analyze preference along the selection procedure for our method and Greedy Coreset (Borsos et al., 2020), and selection experiment is the same in Section 5.1. For both methods, we analyze preference with subset size 25, 50, 75, 100, 125, 150, 175, 200. Results on MNIST, CIFAR-10 and CIFAR-100 are shown in Figure 8, Figure 9 and Figure 10 respectively.

For MNIST dataset, from Figure 8, both our method and Greedy Coreset tend to select harder samples. When subset size smaller than 100, our method selects more hard samples, this explains why our method under-perform Greedy Coreset when subset size is smaller in Figure 1 (a), and as subset size increases, the preference of both methods tend to be the same, this is also consistent to test accuracy in Figure 1 (a). Note that, compared to CIFAR-10 and CIFAR-100, since MNIST is much simpler, both methods tend to select hard-to-learn samples, indicating these samples are discriminative.

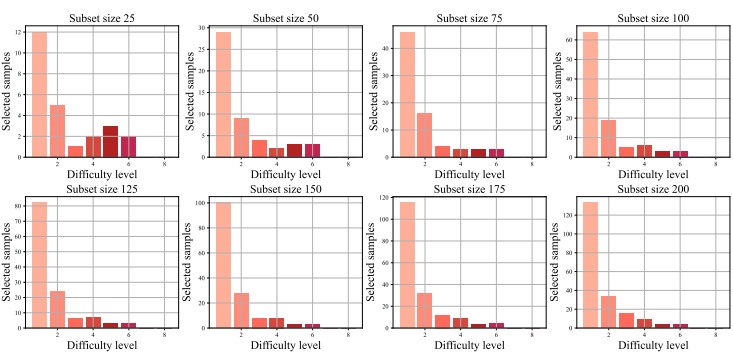

(a) Number of selected sample by our method from each difficulty level during selection on CIFAR-10

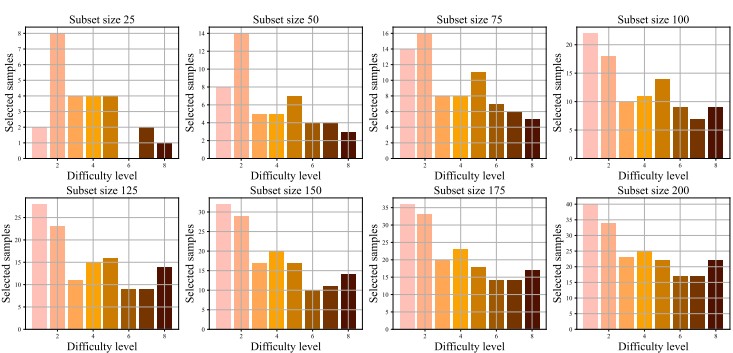

(b) Number of selected sample by Greedy Coreset from each difficulty level during selection on CIFAR-10

Figure 9: Difficulty analysis on CIFAR-10 during selection.

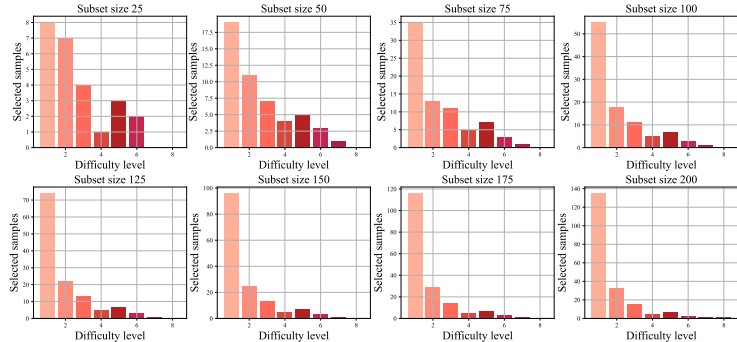

(a) Number of selected sample by our method from each difficulty level during selection on CIFAR-100

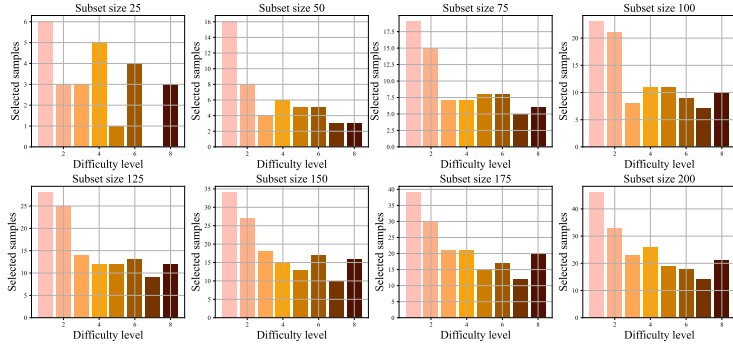

(b) Number of selected sample by Greedy Coreset from each difficulty level during selection on CIFAR-100

Figure 10: Difficulty analysis on CIFAR-100 during selection.

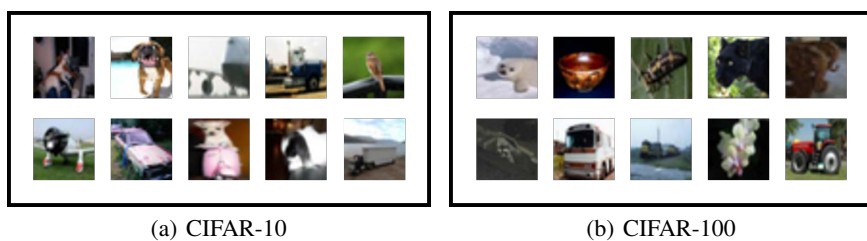

(a) CIFAR-10                    (b) CIFAR-100

Figure 11: Visualization of selected samples by Greedy Coreset in the last two difficulty groups in CIFAR-10 and CIFAR-100.

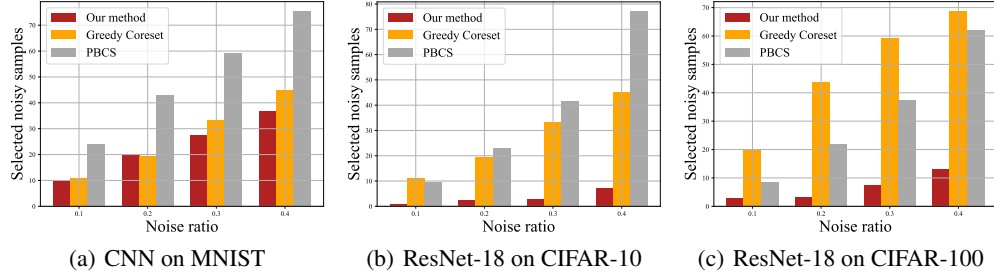

| (a) CNN on MNIST | (b) ResNet-18 on CIFAR-10 | (c) ResNet-18 on CIFAR-100 |

Figure 13: Number of selected noisy samples under different noise ratio. Our method selects much less noisy samples compared to other two methods, demonstrating that our method are more robust under noisy data case.

For CIFAR-10 dataset, from Figure 9, our method tend to select more easy-to-learn samples compared to Greedy Coreset. Note that our method selects no samples from the last two difficulty levels which indicate ambiguous or noisy samples, while Greedy Coreset selects much more samples in last three difficulty levels. We have visualized the samples selected by Greedy Coreset from the last two difficulty levels in Figure 11, and observed that, while these samples are correctly labeled, some samples are non-typical and may contain misleading features that negatively impact the model's learning process. These results demonstrate that CSReL could effectively avoid selecting noisy or ambiguous samples. Experiments in Figure 1 (b) indicate that these harder samples may harm model performance.

For CIFAR-100 dataset, from Figure 10, the results are similar to results on CIFAR-10. Our method selects much less samples in high difficulty levels. According to test performance in Figure 1 (c), our methods selects less ambiguous samples or noisy samples, and perform better than Greedy Coreset.

In conclusion, our method could identify discriminative samples and effectively avoid selecting ambiguous or noisy samples.

## J   ANALYSIS ON REPRESENTING FULL DATASET

Same as experiment in Section 5.3, we also conduct experiments on CIFAR-100 dataset. We train models on selected subset along the selection procedure, and evaluate these models by test accuracy and average loss on full dataset. loss and accuracy with respect to subset size is shown in Figure 12.

From results in Figure 12 model trained on subset selected by our method could achieve higher test accuracy and lower average loss on full dataset. These results demonstrate that, compared to Greedy coreset (Borsos et al., 2020), our method selects a subset with higher performance gain. This finding aligns with the results of our ablation study on the CIFAR-10 dataset, presented in Section 5.3.

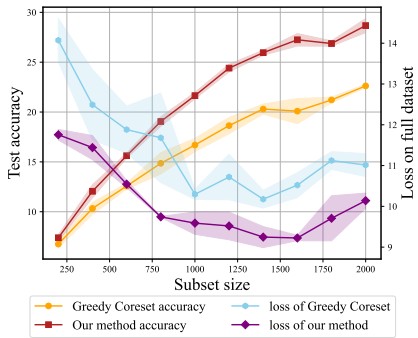

Figure 12: Average loss on the full dataset and test accuracy of models trained on subsets with different size of our method and Greedy Coreset on CIFAR-100 dataset.

## K   DATA SUMMARIZATION UNDER DATA NOISE

We count the number of noisy samples selected by different methods in the experiment of Section 5.3, with results shown in Figure 13. Our method selects significantly fewer noisy samples, particularly on the more challenging CIFAR-10 and CIFAR-100

Table 5: Forgetting of continual learning experiments.

| Methods | Split MNIST | Split CIFAR-10 | Split CIFAR-100 | Perm-MNIST |
|---|---|---|---|---|
| Uniform sampling | 6.26±1.20 | **26.93±4.54** | 79.55±2.19 | 10.29±0.94 |
| $k$-means of features | 6.29±1.42 | 60.98±1.53 | 79.00±1.61 | 10.55±0.63 |
| $k$-center of embeddings | 5.05±1.69 | 56.71±4.22 | 78.82±1.12 | 10.28±0.01 |
| Hardest samples | 15.35±3.16 | 72.55±1.94 | **73.97±1.15** | 11.74±0.71 |
| iCaRL's selection | 5.45±0.32 | 58.64±4.11 | 79.65±1.02 | 9.77±0.26 |
| Greedy Coreset | **3.20±0.49** | 58.41±3.68 | 79.01±0.86 | 9.45±0.23 |
| PBCS | 5.65±0.80 | 53.25±1.61 | 77.00±1.76 | 12.10±1.04 |
| CSReL-CL | **3.03±0.50** | 43.43±5.44 | 75.09±1.24 | **8.81±0.50** |
| CSReL-CL-Prv | 3.92±0.04 | **37.94±8.21** | **72.36±0.15** | **8.44±0.19** |

datasets compared to MNIST. These findings align with the performance drop observed in Section 5.3, demonstrating that our method is more robust in handling noisy data.

## L   FORGETTING IN CONTINUAL LEARNING EXPERIMENTS

In this section, we present forgetting metric (Chaudhry et al., 2018a) which measures performance degradation in subsequent tasks for experiments in Section 5.2. Computation of forgetting is

$$f_j^k = \max_{l \in \{1,...,k-1\}} a_{l,j} - a_{k,j}, \quad \forall j < k,$$

where $a_{k,j}$ denotes accuracy of task $j$ after training $k$-th task. We evaluate forgetting of CL experiments in Table 5.

Among coreset selection methods, our method has the minimal forgetting on all datasets. Our method could also outperform other sample selection methods in most datasets. This demonstrates the effectiveness of our method for selecting a informative subset to prevent forgetting.

## M   CONTINUAL LEARNING ON BCSR SETTING

To further demonstrate the effectiveness of our method, we evaluate CSReL-CL under the same setting as BCSR (Hao et al., 2024) on the Split CIFAR-100 dataset, which is equally divided into 20 tasks. The memory size is set to 100, and task IDs are provided during inference. We replace the selection method in BCSR with our CSReL selection. The results are presented in Table 6, with BCSR and other baseline results referenced from the BCSR paper (Hao et al., 2024).

The results in Table 6 show that our method could outperform other coreset selection baselines, demonstrating the effectiveness of our selection method.

## N   CONTINUAL LEARNING EXPERIMENT ON IMAGENET 500 CLASS

To evaluate the scalability of our CSReL-RS on a large dataset, we select 500 classes from ImageNet-1K and split them into 10 tasks, with the memory size set to 1000. The baselines are ER (Riemer et al., 2018) and DER++ (Buzzega et al., 2020), we replace reservoir sampling in the baseline methods with CSReL-RS as CSReL-ER and CSReL-DER++. For DER++, we use ReL-cmb for selection. The final average accuracy is presented in Table 7.

Results in Table 7 show that replacing reservoir sampling in ER (Riemer et al., 2018) with CSReL-RS leads to a slight performance improvement, while combining our method with DER++ (Buzzega et al., 2020) results in a significant performance boost. These findings demonstrate that our method remains effective on larger datasets, showcasing its scalability for larger datasets.

Table 6: Average accuracy on Split CIFAR-100 under BCSR setting, our CSReL selection could consistently outperform other methods.

| Method | Average accuracy | Forgetting |
|---|---|---|
| k-means features (Nguyen et al., 2017) | 57.82±0.69 | 0.070±0.003 |
| k-means embedding (Sener & Savarese, 2017) | 59.77±0.24 | 0.061±0.001 |
| Uniform | 58.99±0.54 | 0.074±0.004 |
| iCaRL (Rebuffi et al., 2017) | 60.74±0.09 | **0.044±0.026** |
| Grad Matching (Campbell & Broderick, 2019) | 59.17±0.38 | 0.067±0.003 |
| SPR (Kim et al., 2021) | 59.56±0.73 | 0.143±0.064 |
| MetaSP (Sun et al., 2022a) | 60.14±0.25 | 0.056±0.230 |
| Greedy Coreset (Borsos et al., 2020) | 59.39±0.16 | 0.066±0.017 |
| GCR (Tiwari et al., 2022) | 58.73±0.43 | 0.073±0.013 |
| PBCS (Zhou et al., 2022b) | 55.64±2.26 | 0.062±0.001 |
| OCS (Yoon et al., 2021) | 52.57±0.37 | 0.088±0.001 |
| BCSR (Hao et al., 2024) | 61.60±0.14 | 0.051±0.015 |
| **CSReL** | **62.10±0.45** | 0.094±0.006 |

Table 7: Average accuracy on ImageNet 500 class dataset, our method could consistently outperform the random counterpart.

| Method | Average accuracy |
|---|---|
| ER | 9.85±0.01 |
| CSReL-ER | 10.30±0.26 |
| DER++ | 15.94±0.94 |
| **CSReL-DER++** | **19.03±0.21** |

## O    EFFECT OF HOLDOUT MODEL TRAINING EPOCH IN CONTINUAL LEARNING

We also test the effect of holdout model training epoch in CL task. Specifically, we test different holdout model training epochs on Split CIFAR-100 dataset with the same setting as Section 5.2.1. The results are shown in Table 8.

In practice, we find training holdout model for 10 epochs is enough for good performance, compared to 100 training epochs, the computation overhead is small. We show that final results is relatively stable with respect to different holdout model training epochs.

## P    TIME COST AND MODEL SIZE IN DATA SUMMARIZATION EXPERIMENT

For data summarization experiment in Section 5.1, we list time cost and model parameters on MNIST, CIFAR-10 and CIFAR-100 in Table 9. All experiments are conducted on NVIDIA RTX 3090. Specifically, We use 5-layer CNN on MNIST and use ResNet-18 for CIFAR-10 and CIFAR-100, and 200 coreset samples are selected from 10000 candidate samples for each dataset.

Compared to MNIST, selecting corest in CIFAR-10 and CIFAR-100 takes 7x time. However, compared to scaling of model size and data size, the time cost of training holdout model scales mildly.

Table 8: Effect of holdout model training epoch in continual learning.

| Holdout training epochs | Average accuracy |
|---|---|
| 10 | 17.48±0.21 |
| 15 | 17.21±0.07 |
| 20 | 17.43±0.55 |

Table 9: Time cost and model parameters in data summarization experiment.

| Dataset | Model parameters | Holdout training time | Selection time |
|---|---|---|---|
| MNIST | 184.59K | 159.84s | 102.70s |
| CIFAR-10 | 11.16M | 349.74s | 737.68s |
| CIFAR-100 | 11.21M | 350.87s | 735.85s |

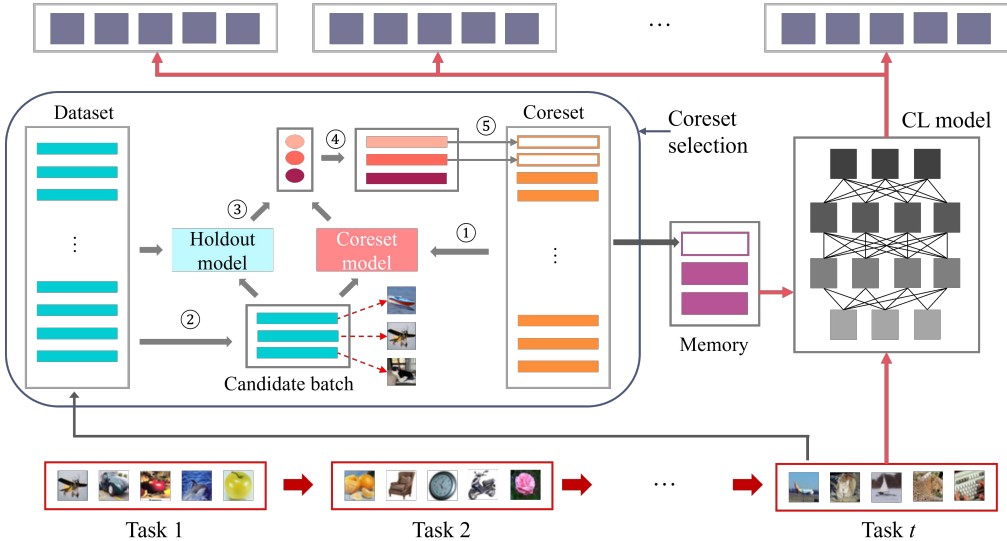

Figure 14: Overview of coreset continual learning process. Coreset is selected from current task dataset after training on current tasks. The selected coreset are added into memory for replay. In coreset selection, holdout model is trained on current task dataset initially, then coreset is selected in multiple steps. In each selection step, ①train coreset model on currently selected coreset, ②select candidate samples, ③compute ReL for each candidate sample, ④rank candidate samples by ReL, ⑤select samples with top-$n$ ReL into coreset.

Besides, time cost of selection also scales mildly as model size increases and data size. Our baseline Greedy Coreset (Borsos et al., 2020) takes 4000 seconds for computing Neural Tangent Kernel, compared to Greedy Coreset (Borsos et al., 2020), our method is much more efficient.

## Q  OVERVIEW OF CORESET SELECTION FOR CONTINUAL LEARNING

To provide a clearer understanding of our method, we further illustrate our CSReL-CL approach, proposed in Section 4.1 in Figure 14. For the continual learning task, we treat the training dataset of the current task as the entire dataset and select the coreset from it. After training the continual learning (CL) model on each task, the holdout model is then trained on the dataset of the current task.

In CSReL-CL-prv proposed in Section 4.2, holdout model is trained on current task dataset and memory. The holdout model is updated after training on each task, rather than being trained only at the initial step. In the CSReL-RS method, introduced in Section 4.3, we use the CL model itself as the holdout model. Consequently, the holdout model incorporates updated information, enabling it to provide valuable insights for the selection process.

## R  IMPACT OF SELECTION ORDER

Since our method selects samples for the coreset using a greedy incremental framework, meaning that once a sample is selected, it is no longer considered in subsequent selections. The problem

of selection order is a common challenge in greedy methods. There are possibility that the performance gain of selected samples might decrease or that some selected samples could be detrimental. To address this, we conducted experiments on MNIST, CIFAR-10 and CIFAR-100 by iteratively removing $n$ samples from the 200-size selected coreset of with the lowest performance gain, then reselecting new samples from the remaining candidates. The results, presented in Table 10, show that performance remains stable after removal and reselection. Additionally, the reselection process produces coresets different from the original ones.

One possible explanation is the existence of multiple subsets containing a similar amount of knowledge. Our method selects samples that are both representative and complementary to the current coreset. This implies that selecting different initial samples results in similar final performance because our method inherently identifies subsets that complement the initial selection. Further evidence for this comes from our evaluation of the coreset selection method under different random seeds in Section 5.1. Despite the variation in selected subsets, the performance remains consistent. Therefore, the order of selection is not a significant concern in real-world data scenarios.

Table 10: Test accuracy after removing and reselecting samples for coreset.

| $n$ | MNIST | CIFAR-10 | CIFAR-100 |
|---|---|---|---|
| 0 | 96.67 | 39.46 | 9.26 |
| 10 | 96.53 | 39.58 | 10.01 |
| 20 | 96.51 | 39.60 | 8.93 |
| 50 | 96.42 | 39.16 | 10.08 |
| 100 | 96.09 | 39.31 | 10.19 |

## S    DISTINGUISHING DIFFICULT SAMPLES FROM NOISY SAMPLES

We note that difficult samples and noisy samples cannot be distinguished solely based on Eq. (4) and (6), as both types of samples may exhibit high loss values on the holdout model. However, our method selects coreset iteratively. If noisy or ambiguous samples are included in the coreset, they may hinder the coreset model $\theta_S$ from effectively learning other clean samples, leading to increased loss for the clean candidates on the coreset model. As a result, the ReL of clean candidates may increase, making them more likely to be selected in subsequent iterations.

While difficult samples could be informative and not harm the learning of other samples, selecting difficult samples does not preclude the selection of other difficult samples, as demonstrated in Figure 8(a) in Appendix I. For the relatively simple MNIST dataset, difficult samples tend to be more informative and are therefore selected. Additionally, we analyzed the difficulty levels of selected samples in the noisy MNIST experiment discussed in the Ablation Study (Section 5.3) in Figure 15. The results show that more easy samples are selected from the noisy MNIST dataset, due to the learning of these samples is disrupted by the presence of selected noisy samples. Consequently, difficult samples and noisy samples yield different outcomes. Additionally, we note that the noise rates in the last two difficulty groups shown in Figure 15 are 0.24% and 79.76% respectively. Our method tends to select more samples from the second-to-last group while selecting fewer samples from the last group. This is because the holdout model can effectively learn difficult samples, whereas noisy samples cannot be well learned by either the holdout model or the coreset model.

In summary, our method is capable of selecting more difficult samples when they are informative and consistent with other samples. For noisy and ambiguous samples, however, our approach prioritizes cleaner samples to mitigate the disruption caused by noisy data. In low noise-ratio scenarios, which are common in real-world applications, noisy samples can be identified in as the holdout model can effectively learn difficult samples but fails to learn noisy ones. As a result, our method effectively treat difficult and non-representative samples, ensuring appropriate selection.

## T    CORESET SELECTION FOR CONTINUAL LEARNING WITH LARGE
## BACKBONE MODELS

To further demonstrate the effectiveness and efficiency of our method over other bilevel coreset selection methods in continual learning. We conducted additional experiments applying complex backbones ResNet-50 and VIT-Tiny (Wu et al., 2022) on CIFAR-100 and Tiny-ImageNet respec-

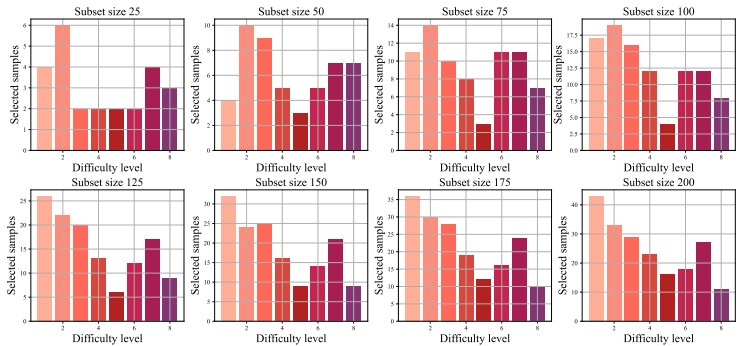

Figure 15: Number of selected samples from each difficulty group in noisy MNIST experiment, noise-ratio=0.1.

Table 11: Average accuracy and time cost of Greedy Coreset, BCSR and CSReL-CL on CIFAR-100 and Tiny-ImageNet dataset.

|  | CIFAR-100 | | Tiny-ImageNet | |
|---|---|---|---|---|
|  | Average accuracy | Time cost | Average accuracy | Time cost |
| Greedy Coreset | 28.17±0.57 | 21h 58m | 12.17±0.07 | 24h 55m |
| BCSR | 27.32±0.10 | 11h 26m | 12.33±0.08 | 28h 13m |
| **Our method** | **32.51±0.58** | **10h 04m** | **17.51±0.11** | **15h 13m** |

tively. Both datasets were evenly split into 10 tasks, with memory sizes set to 1000 and 2000 for CIFAR-100 and Tiny-ImageNet, respectively.

We compared our method with related bilevel coreset selection methods for continual learning, including Greedy Coreset (Borsos et al., 2020) and BCSR (Hao et al., 2024). The average accuracy and time cost are summarized in Table 11. Our method outperforms other bilevel coreset selection approaches by a large margin on both datasets while also being more efficient, particularly when using the more complex VIT-Tiny backbone.

## U  FEATURE MAP OF SELECTED SAMPLES

We have visualized the features of coreset samples selected by different methods from CIFAR-10 using t-SNE. The features were extracted using a ResNet-18 model trained on CIFAR-10, as shown in Figure 16. Representative samples share common features with other samples and are effectively learned by the model, thus could be well classified. The features of the coreset selected by our method are concentrated near the high density part and better separated compared to those selected by the two baseline methods, demonstrating the ability of our approach to select representative samples.

Our work defines informative sample as samples contains new knowledge compared to currently selected coreset. To demonstrate that our method selects both representative and informative samples, we train model on the first 150 selected samples and visualize their features as dots. We then visualize the next 50 selected samples as triangles. Figure 17 shows the features of coreset samples selected by our method and by Greedy Coreset. The later-selected samples by our method tend to lie near the margins of feature clusters, indicating they are less well-learned by the model. Furthermore, samples selected by our method are more closely aligned with the feature clusters of each class, while those selected by Greedy Coreset are more scattered. This demonstrates that our method effectively selects samples that are both representative and informative. However, it is important to note that the informativeness of a sample cannot be determined solely based on its features, as not

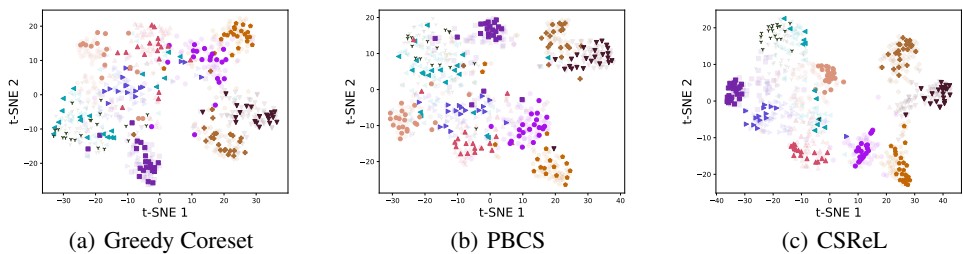

(a) Greedy Coreset       (b) PBCS       (c) CSReL

Figure 16: Features of selected samples by Greedy Coreset (Borsos et al., 2020), PBCS (Zhou et al., 2022b) and CSReL. The shadow part is features of randomly selected samples from the full dataset. Our method selects more representative samples compared to other methods.

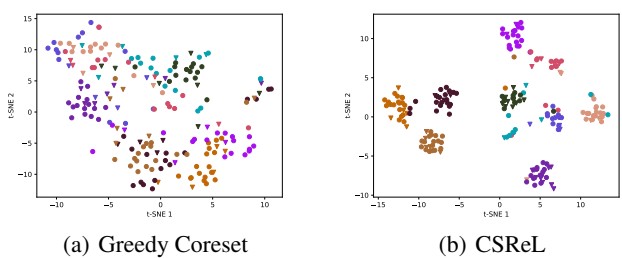

(a) Greedy Coreset       (b) CSReL

Figure 17: Features of currently selected samples and later selected samples, illustrating that our method selects both representative and informative samples.

well learned samples, such as ambiguous or noisy ones, may negatively impact the model's learning process.

We have also plotted the features of samples selected by iCaRL's selection (Rebuffi et al., 2017) and $k$-means features selection (Nguyen et al., 2017) in Figure 18. Compared to Greedy Coreset Borsos et al. (2020) and PBCS (Zhou et al., 2022b), features are less scattered.

## V  ADDITIONAL DATA SUMMARIZATION RESULTS

We further conduct coreset selection experiments with RCS (Xu et al., 2023) and CDS (Wan et al., 2024) on the CIFAR-10 and CIFAR-100 datasets under the same settings as Section 5.1, with the coreset size set to 200. The results are presented in Table 12.

RCS (Xu et al., 2023) defines a bilevel selection objective for adversarial contrastive learning, which differs from the supervised classification task addressed in our work. As a result, RCS underper-

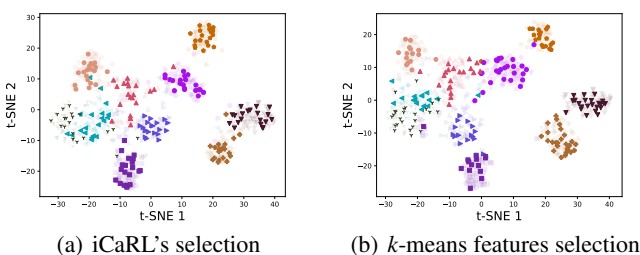

(a) iCaRL's selection       (b) $k$-means features selection

Figure 18: Features of samples selected by iCaRL's selection and $k$-means features selection.

Table 12: Test accuracy of model trained on coreset selected by different methods, our method outperforms all other baseline methods.

| Method | CIFAR-10 | CIFAR-100 |
|---|---|---|
| RCS | 27.17±1.11 | 4.87±0.60 |
| GC+CDS | 33.03±1.74 | 8.06±0.23 |
| Greedy Coreset | 36.53±1.40 | 6.94±0.23 |
| PBCS | 37.41±0.26 | 6.77±0.18 |
| **CSReL (ours)** | **39.58±0.14** | **8.91±0.42** |

forms other coreset selection methods. GC+CDS (Wan et al., 2024) leverages the contributing dimension structure within the Graph Cut framework (Iyer et al., 2021), constructing coresets based on a balance between representativeness and diversity. However, unlike bilevel optimization methods, this approach relies on hyperparameter tuning to achieve this balance, which may be suboptimal. Additionally, the improvement in model performance when adding a sample to the coreset cannot be directly evaluated solely through representativeness and diversity metrics. In contrast, bilevel optimization methods select samples by evaluating performance in the outer loop, thereby providing stronger performance guarantees for the resulting coreset.

bilevel coreset selection methods outperform RCS (Xu et al., 2023) and GC+CDS (Wan et al., 2024) on the CIFAR-10 dataset and perform slightly below GC+CDS (Wan et al., 2024) on the CIFAR-100 dataset. These results indicate that bilevel coreset selection methods remain effective and robust. Furthermore, our CSReL selection method surpasses all baseline methods, showcasing its superior effectiveness and timeliness.

## W    DATA SUMMARIZATION EXPERIMENT DETAILS

We conduct experiments on MNIST, CIFAR-10 and CIFAR-100, coreset size of all datasets is set to 200. For experiments on MNIST and CIFAR-10, we follow experiment settings of Borsos et al. (2020) and Zhou et al. (2022b). Different from Greedy Coreset which uses Neural Tangent Kernel (Jacot et al., 2018) as inner model, we use same model structure for holdout model and current model.

We use CNN as backbone for MNIST, which contains two blocks of convolution, dropout, maxpooling and ReLU activation, two convolution layers have 32 and 64 filters with $5 \times 5$ kernel size. Two fully connected layers of size 128 and 10 with dropout follows convolution blocks. The dropout probability is 0.5. We train CNN on selected coreset using SGD optimizer with learning rate $2e^{-2}$, training batch size is set to 32 and training epochs is set to 3000. This training protocol is applied in all compared methods. The reason why we don't use Adam optimizer is that we found performance is not stable under different random seeds, therefore, we use SGD optimizer instead.

We use same ResNet-18 as in Borsos et al. (2020) and Zhou et al. (2022b) as backbone for CIFAR-10 and CIFAR-100 dataset. Note that there is no Batch-Normalization layer in this backbone. For experiments on CIFAR-10 and CIFAR-100, we train ResNet on selected coreset using Adam optimizer with learning rate $5e^{-4}$, training batch size is set to 64 and training epochs is set to 1800. This training protocol is applied in all compared methods.

In experiment on MNIST, holdout model is trained by SGD optimizer with learning rate $1.5e^{-3}$, training batch size is 32 and training epoch is 125. In each selection step, current model is trained by SGD optimizer with learning rate $2e^{-2}$, batch size is set to 32. Since selected coreset is very small, to avoid overfitting ,we train current model on selected subset with 16 epochs. Initial coreset size is set to 0 and selection step is 200.

In experiment on CIFAR-10 and CIFAR-100, holdout model is trained by SGD optimizer with learning rate $3e^{-3}$, batch size is set to 32 and training epoch is 100. In each selection step, current model is trained by SGD optimizer with learning rate $2e^{-2}$, batch size is set to 32, current model training epoch is set to 16. Initial coreset size is set to 0 and selection step is 200.

Table 13: Optimization hyperparameters in continual learning experiments in Table 1.

| Dataset | Batch size | Epochs | Optimizer | Learning rate | Loss factor |
|---|---|---|---|---|---|
| Split MNIST | 256 | 400 | Adam | 5e-4 | 100.0 |
| Split CIFAR-10 | 256 | 400 | Adam | 5e-4 | 20.0 |
| Split CIFAR-100 | 32 | 100 | SGD | 2e-2 | 4.0 |
| Permuted MNIST | 256 | 400 | Adam | 5e-4 | 0.1 |

Table 14: Optimization hyperparameters in continual learning experiments in Table 2.

| Dataset | Batch size | Optimizer | Learning rate | CE Loss factor | KD loss factor |
|---|---|---|---|---|---|
| Split CIFAR-100 | 32 | SGD | 2e-2 | 1.0 | 0.2 |
| Split Tiny-ImageNet | 32 | SGD | 3e-2 | 1.0 | 0.1 |

## X  CONTINUAL LEARNING EXPERIMENT DETAILS

### X.1  DATA SUMMARIZATION FOR CONTINUAL LEARNING

**Datasets:** We conduct experiments on Split MNIST, Split CIFAR-10, Split CIFAR-100 and Perm MNIST. Split MNIST, CIFAR-10 and MNIST consist of 10 classes. Following Borsos et al. (2020) we split CIFAR-10 and MNIST into 5 tasks with 2 classes for each task. We split CIFAR-100 into 10 tasks with 10 classes for each class. For Perm MNIST contains 10 tasks and each task is randomly permuted version of MNIST. Also following Borsos et al. (2020), in experiments in Split MNIST, Split CIFAR-10 and Perm MNIST, we randomly select 1000 samples from each task for training, for Split CIFAR-100, we use all data of each task.

**Augmentations:** Following Borsos et al. (2020), we apply normalization to Split MNIST and Perm MNIST. For more complicated CIFAR-10 and CIFAR-100 dataset, we apply random crop, random horizontal flip and normalization.

**Backbones:** We use same CNN structure in Section W for Split MNIST, and use the same ResNet-18 in Section W for Split CIFAR-10. For Split CIFAR-100, we add BatchNormalization layer in each convolution block of ResNet-18. For Perm MNIST, we use a fully connected net with two hidden layers with 100 units, ReLU activations, and dropout with probability 0.2 on the hidden layers.

**Optimization:** Optimization related hyperparameters are shown in Table 13, the loss factor is $\alpha$ in Eq. (17). Following Borsos et al. (2020) and Zhou et al. (2022b), we use Adam optimizer (Kingma, 2014) for experiments on Split MNIST, Split CIFAR-10 and Permuted MNIST. For Split CIFAR-100 dataset, we use SGD optimizer for all compared methods.

### X.2  CONTINUAL LEARNING WITH MODIFIED RESERVOIR SAMPLING AND KNOWLEDGE DISTILLATION

**Datasets:** We conduct experiments on Split CIFAR-100 and Split Tiny-ImageNet dataset. We split CIFAR-100 into 10 tasks with 10 classes for each class. For Split Tiny-ImageNet, we split totally 200 classes into 10 disjoint tasks. All samples of each task are used during training.

**Augmentations:** We keep the same augmentation used in mammoth (Buzzega et al., 2020). Both Split CIFAR-100 and Split Tiny-ImageNet dataset applies random crop, random horizontal flip and normalization.

**Backbones:** Both dataset use the same ResNet-18 in mammoth implementation (Buzzega et al., 2020).

**Optimization:** We list all the hyperparameters related to optimization in Table 14. Training epoch is consistent with corresponding baselines.

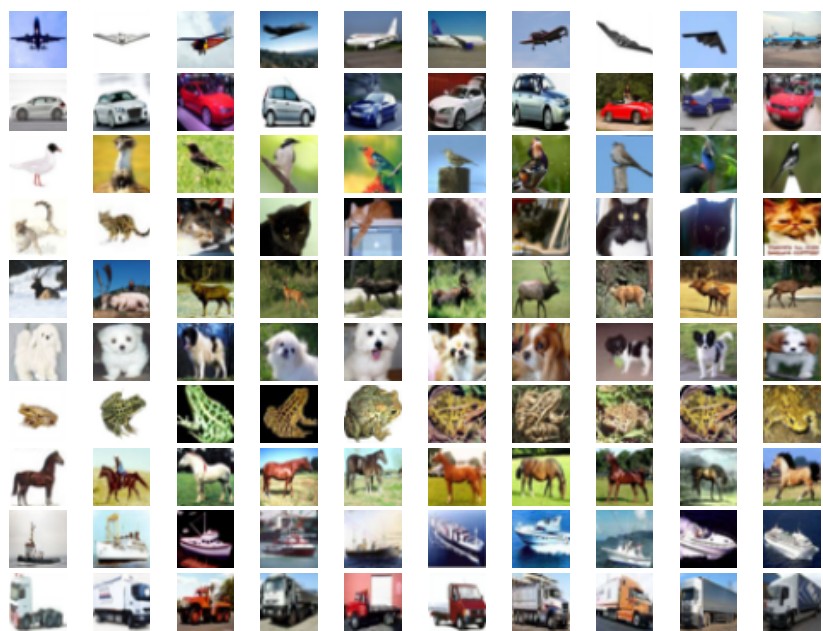

Figure 19: Visualization of selected sample in each class from CIFAR-10.

Table 15: Time cost for offline continual learning tasks

|  | Split MNIST | Split CIFAR-10 | Perm MNIST | Split CIFAR-100 |
|---|---|---|---|---|
| Time cost | 4.8m | 27.6m | 17.5m | 69.4m |

## Y    VISUALIZATION OF SELECTED DATA

To make the effectiveness of our method straight forward, we visualize the first 10 selected samples of each class in CIFAR-10, pictures are shown in Figure 19, each row collects images from same class.

From visualization, we can see that sample selected by our method is clear, unambiguous and diverse.

## Z    COMPUTATION RESOURCES AND EXPERIMENT TIME COST

We conduct all experiments on 2 NVIDIA GeForce RTX 3090 graphical cards with 24 GB memory for each graphical card. Our CPU type is Intel(R) Core(TM) i9-12900K, memory of our server is 32 GB. We list time cost for all main offline continual learning experiments in Table 15 and list time cost for CSReL-RS in Table 16.

Table 16: Time cost for continual learning tasks using CSReL-RS

|  | Split CIFAR-100 | Split Tiny ImageNet |
|---|---|---|
| CSReL-ER | 56m | 7h 47m |
| CSReL-DER++ | 1h 20m | 8h 56m |

## AA URL of Cited Assets

We download MNIST dataset from http://yann.lecun.com/exdb/mnist/.

We download CIFAR-100 dataset from https://www.cs.toronto.edu/ kriz/cifar-10-python.tar.gz.

We download CIFAR-100 dataset from https://www.cs.toronto.edu/ kriz/cifar-100-python.tar.gz.

Tiny ImageNet dataset is downloaded by mammoth implementation, the URL is https://github.com/aimagelab/mammoth.

Greedy Coreset implementation is downloaded from https://github.com/zalanborsos/bilevel_coresets.

## AB Broader impact

Coreset selection aims to summarize an informative subset from full dataset, which could significantly reduce training cost and energy consumption. Storage required may also be reduced with coreset selection method. Our coreset selection method could be applied not ony on continual learning, but also Neural Architecture Search and Reinforcement learning. Continual learning in our work could also improve intelligent agent for continually learning new knowledge, enabling more general applied models and personalized models.

