# OpenReview forum: "Coreset Selection via Reducible Loss in Continual Learning"
_ICLR.cc/2025/Conference — ICLR 2025 Poster_

### Official Review · Reviewer_Ungg · 2024-11-03

**Soundness:** 2
**Presentation:** 3
**Contribution:** 2
**Rating:** 5
**Confidence:** 5

**Summary:**

The authors focus on the coreset selection method used in continual learning scenario. They believe that the not well learned samples, like ambiguous and noise samples should not be included in the coreset. Therefore, a new method CSRL is designed with reducible loss to filter the not-well-learned samples during coreset selection. The experiments are conducted on MNIST, CIFAR-10, and CIFAR-100 with several earlier competitor methods.

**Strengths:**

1. I find the authors make great efforts in polishing this work. I found multiple supplementary explanations for the main text in the supplementary materials, which I believe is commendable and should be encouraged.
2. The paper is well-organized and writing is almost clear.

**Weaknesses:**

About Method:

1. The proposed CSRL seems to be severely affected by the selection order and subset division way. If Algorithm 1 is correctly implemented, this means that the selected samples in the coreset will not be considered in the following selection process. Obviously, the gain score G of samples stored in the coreset will change as the model gradually learns from all subsets. The authors ignore this possibility, which may lead to the result that previously selected candidates are more likely to be stored in the coreset. In other words, the gain score may not be fair for all candidates.

2. The gain score G seems only to assess which samples are easier to learn fully, rather than which are more core. From the expressions in Equations (4) to (6), it is evident that the authors do not differentiate between difficult samples and noise when applying this gain score, instead treating both as noise. Clearly, difficult samples should also be treated appropriately during coreset selection.

3. I notice that the authors provide the results in Appendix H. However, I believe these results do not verify that CSRL is more reasonable and may even exacerbate the concerns I raised in W1. Since the overall difficulty level of the entire dataset is not provided, we cannot determine whether CSRL regulates the coreset selection more effectively. As I expressed in W1, the results can also be interpreted to mean that CSRL is severely affected by the selection order (which seems to be random) during application, leading to selection results that might also be random as the size of the coreset increases. This suggests that CSRL has less impact on regulating coreset selection. In contrast, the other competing method appears to perform better. Additionally, the authors claim that selecting fewer hard-to-learn samples will make a difference, which further confirms that hard samples are considered the same as noise samples.


About Experiments:

4. I noticed that the settings used by the authors are somewhat outdated. First, the largest network structure in this paper is only ResNet-18, and the authors further provide training and selecting time results based on this backbone to show the efficiency of their method. From both the performance and efficiency perspectives, I find this unconvincing. With such simple network structures, it is difficult to compare performance differences between new and old methods. Let alone that many recent state-of-the-art methods are not included in the main experimental results.

5. In terms of the experimental results, I find them unconvincing because the authors only adopt competitor methods from 3 and 5 years ago in all the experimental results presented in the main text. It becomes even more puzzling when I notice that the authors are aware of the latest state-of-the-art methods (e.g., BCSR, GCR) but only compare CSRL with them on one of the datasets used in BCSR in the appendix. Why are the latest state-of-the-art methods not compared under the same settings in this paper's main text? I believe that both the absence of state-of-the-art methods in the main text and the partial comparison results of these methods shown only in the appendix do not support the authors' claim that the proposed CSRL is state-of-the-art.

**Questions:**

Please see my questions in Weaknesses.

---

> ### Author Response · Authors · 2024-11-21
> **Response to Weakness 1-2**
>
> Thank you for your detailed review and thoughtful feedback. We appreciate your recognition of the effort put into our work and your constructive comments on the methodology and experimental setup. Your insights have been valuable in identifying areas for improvement, and we have addressed your concerns in our responses.
>
> **Weakness**
>
> 1. This is an excellent question. The issue of selection order is a common challenge in greedy methods, which are employed due to the NP-hard nature of the coreset selection problem. Our method selects samples for the coreset using a greedy incremental framework, where once a sample is selected, it is excluded from consideration in subsequent selections. We have addressed this problem as follows:
>    - We previously considered the possibility that the performance gain of selected samples might decrease or that some selected samples could be detrimental, and conducted experiments on MNIST, CIFAR-10 and CIFAR-100 by iteratively removing $n$ samples from the selected coreset with the lowest performance gain, then reselecting new samples from the remaining candidates. The results, presented in Table below, show that performance remains stable after removal and reselection. Additionally, the reselection process produces coresets different from the original ones.
>    - One possible explanation is the existence of multiple subsets containing a similar amount of knowledge.
>    - Our method selects samples that are both representative and complementary to the current coreset. This implies that selecting different initial samples results in similar final performance because our method inherently identifies subsets that complement the initial selection.
>    - Further evidence for this comes from our evaluation of the coreset selection method under different random seeds in Section 5.1. Despite the variation in selected subsets, the performance remains consistent.
>    - Therefore, the order of selection is not a significant concern in real-world data scenarios. We have also added this discussion in Appendix Q. in our updated submission.
>
>
> | $n$ | MNIST | CIFAR-10 | CIFAR-100 |
> |:---:|:-----:|:-----:|:-----:|
> | 0   | 96.67 | 39.46 | 9.26  |
> | 10  | 96.53 | 39.58 | 10.01 |
> | 20  | 96.51 | 39.60 | 8.93  |
> | 50  | 96.42 | 39.16 | 10.08 |
> | 100 | 96.09 | 39.31 | 10.19 |
>
>
> 2. We acknowledge that difficult samples and noisy samples cannot be distinguished solely based on Equations (4) and (6), as both types of samples may exhibit high loss values on the holdout model. Our method could differentiate difficult and noisy samples in the following aspects.
>     - Our method selects coresets iteratively, difficult and noisy samples may have different impacts during the selection process.
>     - Difficult samples could be informative and not harm the learning of other samples, selecting difficult samples does not preclude the selection of other difficult samples. As demonstrated in Figure 8(a) in Appendix H. For the relatively simple MNIST dataset, difficult samples tend to be more informative and are therefore selected.
>     -  While if noisy or ambiguous samples are included in the coreset, they may hinder the coreset model $\boldsymbol{\theta}_{\mathcal{S}}$ from effectively learning other clean samples, leading to increased loss for the clean candidates on the coreset model. As a result, the CSRL of clean candidates may increase, making them more likely to be selected in subsequent iterations.
>     -  We additionally analyzed the difficulty levels of selected samples in the noisy MNIST experiment discussed in the Ablation Study (Section 5.3). The results show that more easier samples are selected from the noisy MNIST dataset. This indicates that the model prioritizes easier samples when the learning of these samples is disrupted by the presence of selected noisy samples.
>     -  In low noise-ratio cases, which are common in real world applications, we observe that our method can distinguish difficult samples from noisy ones, as demonstrated in the noisy MNIST experiments (noise-ratio = 0.1). This is because the holdout model can effectively learn difficult samples, whereas noisy samples cannot be learned by either the holdout model or the coreset model. In high noise-ratio scenarios, the holdout model is significantly affected by noisy samples and may struggle to effectively learn difficult samples. Distinguishing noisy samples from difficult ones in noisy datasets remains a common challenge, and we plan to address this issue in our future work.

---

> > ### Author Response · Authors · 2024-11-21
> > **Response to Weakness 3-5**
> >
> > **Weakness**
> >
> > 3. Randomness and distinguishing difficult samples from noisy samples is a common challenge in coreset selection, and our method could effective regulate coreset selection in the following aspects:
> >     - As addressed in our responses to Weakness 1 and Weakness 2, our method selects samples with complementary knowledge and does not exclude difficult samples from selection.
> >     - There is some randomness in our selection process, as multiple subsets can provide similar knowledge. However, as shown in Section 5.1, our method significantly outperforms random uniform selection, indicating its effectiveness in regulating coreset selection.
> >     - Greedy Coreset tends to select hard-to-learn samples due to their large loss values, which may result in the inclusion of ambiguous or noisy samples in the coreset. The results in Appendix H. show that more difficult samples are included in the coreset when using Greedy Coreset. We visualized the samples selected by Greedy Coreset from the last two difficulty groups in Figure 11 (Appendix H.) and observed that, while these samples are correctly labeled, some samples are non-typical and may contain misleading features that negatively impact the model's learning process.
> >     - In contrast, the coreset selected by our method includes fewer samples from these difficulty groups since these samples cannot be well learned by holdout model and have lower CSRL. We also note that easier samples can be informative, as they are clear and share common features with other samples. In conclusion, our approach effectively regulates coreset selection.
> >
> > 4. To ensure a fair comparison, we used the same ResNet-18 backbone as the baseline methods. To address your concern on the effectiveness and efficiency of our approach with complex backbones, we conducted additional experiments on CIFAR-100 using a ResNet-50 backbone and Tiny-ImageNet using a VIT-Tiny backbone. Both datasets were evenly split into 10 tasks, with memory sizes set to 1000 and 2000 for CIFAR-100 and Tiny-ImageNet, respectively.
> > &ensp;&ensp; We compared our method with related bi-level coreset selection methods for continual learning, including Greedy Coreset and BCSR. The average accuracy and time cost are summarized in Table below. Our method outperforms other bi-level coreset selection approaches by a large margin on both datasets while also being more efficient, particularly when using the more complex VIT-Tiny backbone. We have also added this comparison in Appendix S. in our updated submission.
> >
> > Average accuracy and time cost on CIFAR-100
> >
> > |                | Average accuracy | Time cost |
> > |----------------|:----------------:|:---------:|
> > | Greedy Coreset | 28.17±0.57 | 21h 58m |
> > | BCSR           | 27.32±0.10 | 11h 26m |
> > | Our method     | 32.51±0.58 | 10h 04m |
> >
> > Average accuracy and time cost on Tiny-ImageNet
> >
> > |                | Average accuracy | Time cost |
> > |----------------|:----------------:|:---------:|
> > | Greedy Coreset | 12.17±0.07 | 24h 55m |
> > | BCSR           | 12.33±0.08 | 28h 13m |
> > | Our method     | 17.51±0.11 | 15h 13m |
> >
> > 5. Since the experimental setting of BCSR differs from that of other works (in terms of backbone, task split, and the availability of task IDs during inference), we applied our method to the BCSR setting in the appendix to ensure a fair comparison. In the main text, we compare our method with other baseline approaches. Additionally, we included experiments for BCSR, GCR, and OCS under the same setting in Table 1 in our updated submission. The results demonstrate that our method outperforms recent coreset selection baselines, highlighting its effectiveness.
> >
> > If you have any further concerns, please feel free to let us know.

---

> > > ### Author Response · Authors · 2024-12-02
> > > **We look forward to your follow-up**
> > >
> > > Dear Reviewer Ungg,
> > >
> > > We hope this message finds you well. We are writing to kindly follow up on the rebuttal we submitted for your review. We have provided detailed responses to your comments and would be happy to clarify or expand on any points if needed.
> > >
> > > We understand that the review process can be quite busy, and we sincerely appreciate the time and effort you’ve dedicated to evaluating our submission. Please let us know if there are any additional questions or concerns we can address during this discussion phase.
> > >
> > > Thank you again for your thoughtful feedback and for helping us improve our work.
> > >
> > > Best regards,
> > > Authors of Submission 1231

---

> > > > ### Comment · Reviewer_Ungg · 2024-12-02
> > > > **Response to authors' rebuttal**
> > > >
> > > > Thanks for the replies with further analysis and results. Some of my concerns have been addressed like the effects of selection order and subset division way. The remaining concern mainly lies in the argument regarding CSRL’s ability to distinguish between difficult and noisy samples. I still insist that current results in Appendix H is not convincing enough to address this issue and suggest that the authors provide the results of the difficulty distribution of each dataset evaluated by the hold-out model. This would help to check that the performance improvement and noise robustness of CSRL are not merely achieved by focusing more on the majority samples in the original data (e.g., simple samples in MNIST and difficult samples in CIFAR100).
> > > >
> > > > In addition, I noticed some potential issues with the newly added comparison methods. For instance, in Table 1, the results on the CIFAR-100 dataset show a significant different superiority over GCR and BCSR compared to that in Table 11, which seems unlikely to be caused solely by changing the backbone. Additionally, the time cost of BCSR increases significantly across datasets in Table 11, which also seems abnormal. I recommend the authors carefully review the implementation and results for these methods.
> > > >
> > > > Overall, considering the above concerns, I keep the rating unchanged.

---

> ### Author Response · Authors · 2024-12-02
> **Response to further concerns**
>
> Thank you for your thoughtful follow-up and for acknowledging the aspects of your concerns that have been addressed, such as the effects of selection order and subset division methods. Regarding your remaining concerns:
>
> 1. We have demonstrated in Appendix R that our method can distinguish between difficult and noisy samples under a low noise ratio. We apologize for not emphasizing this in our rebuttal. Selecting difficult samples versus noisy samples leads to different outcomes, as shown in our analysis in Appendix R. However, we also acknowledge that our method struggles to differentiate between difficult and noisy samples under high noise ratios, which remains a common challenge in coreset selection tasks.
> 2. In Appendix H, we evenly split the samples into eight groups based on the holdout model loss, and there is no majority group in our split. To reflect the difficulty distribution, we provide the average loss of each difficulty group in the table below. Our method selects samples that are representative and contribute new knowledge relative to the currently selected coreset. The selected samples from each difficulty group in Appendix H. represent the outcome of our selection process, not the reasoning behind it.
>
> |         | MNIST | CIFAR-10 | CIFAR-100 |
> |---------|:-----:|:--------:|:---------:|
> |Group 1 | 0.0002 | 0.03     | 0.09      |
> |Group 2 | 0.0008 | 0.12     | 0.47      |
> |Group 3 | 0.0020 | 0.31     | 1.02      |
> |Group 4 | 0.0038 | 0.56     | 1.64      |
> |Group 5 | 0.0072 | 0.87     | 2.26      |
> |Group 6 | 0.0141 | 1.26     | 2.91      |
> |Group 7 | 0.0316 | 1.85     | 3.74      |
> |Group 8 | 0.1565 | 3.31     | 5.30      |
>
> 3. In Table 11, we compared our method with BCSR and Greedy Coreset, which show similar performance on the CIFAR-100 dataset as reported in Table 1. While GCR is not included in Table 11. We believe the results in Table 1 (Greedy Coreset: 15.04\%, BCSR: 15.11\%) and Table 11 (Greedy Coreset: 28.17\%, BCSR: 27.32\%) are consistent.
> 4. Regarding the time cost of BCSR, this method selects candidate samples from each batch and then selects the coreset for memory from these candidates. Since Tiny-ImageNet contains 100,000 samples—twice as many as CIFAR-100, and the number of batches also doubles. Consequently, the selection time doubles as selection step is doubled, which explains why BCSR incurs a higher time cost compared to the other two methods.
>
> We kindly ask that you reconsider the soundness of our work. If there are any remaining clarifications or follow-up questions, we would be happy to address them promptly.

---

> ### Author Response · Authors · 2024-12-04
> **Final clarification on your concern about distinguish between noisy and difficult samples**
>
> Dear reviewer Ungg,
>
> Thank you for your thoughtful comments, which have significantly improved our work, and for your time and effort in reviewing our paper. We would like to provide a final clarification regarding your concern about distinguishing between noisy and difficult samples.
>
> Our method can distinguish these two types of samples in low noise ratios. The holdout model can learn difficult samples, while the model trained on the currently selected subset may exhibit high loss on these samples, resulting in a large CSRL score for difficult samples. In contrast, noisy samples cannot be effectively learned by either model, leading to a low CSRL score and making them less likely to be selected.
>
> Additionally, our method selects samples iteratively, and the inclusion of noisy samples would lead to different outcomes. Noisy samples can negatively impact the learning of other clean samples, resulting in higher loss for those clean samples on the model trained with the current subset. This, in turn, increases the CSRL score of this clean samples, making them more likely to be selected, as discussed in Appendix R. Therefore, our method demonstrates robustness to noisy conditions, as shown in the ablation study (Section 5.3). We apologize for not emphasizing this point sufficiently in our rebuttal.
>
> Separating noisy samples from difficult samples remains a common challenge in coreset selection tasks. As shown in Figure 8(a) (Appendix H), our method does not simply select easy samples. Furthermore, we have demonstrated that our method is more robust under noisy conditions compared to other bi-level coreset selection methods, enhancing both the soundness and effectiveness of our approach.
>
> We hope this response adequately addresses your concern and remain open to any further questions.
>
> Best regards,\
> Authors of submission 1231.

---

### Official Review · Reviewer_kUqK · 2024-11-03

**Soundness:** 2
**Presentation:** 3
**Contribution:** 2
**Rating:** 5
**Confidence:** 5

**Summary:**

This paper introduces a method for coreset selection using reducible loss to improve continual learning (CL) performance. Traditional experience replay-based CL methods face challenges when selecting representative samples from non-stationary data streams, often retaining ambiguous or noisy data that can degrade model performance. The proposed method uses reducible loss as a selection criterion, estimating each sample's performance gain to determine its representativeness and informativeness. This selection framework is computationally efficient, requiring only a forward computation without gradient backpropagation, thus avoiding the computational cost associated with implicit gradient methods. Additionally, the approach is extended to handle challenges in CL, including task interference and data streaming, demonstrating robustness against noisy data and effective scalability for large datasets like ImageNet. Experimental results show that this method consistently improves CL model accuracy and reduces forgetting when applied to various datasets.

**Strengths:**

This paper introduces a coreset selection method based on reducible loss for continual learning, aiming to improve model performance by selecting representative samples efficiently. By focusing on performance gain, this approach reduces computational demands and mitigates the impact of noisy data. It addresses key continual learning challenges, such as task interference and data streaming, making it particularly valuable for dynamic and large-scale applications.

**Weaknesses:**

The coreset selection method based on reducible loss presented in this paper shows promise for continual learning applications, but some of the design choices lack thorough validation, and certain aspects of the experimental comparisons and theoretical evidence could be more comprehensive.

**Questions:**

1.	This paper builds upon the limitations identified in Greedy Coreset (NIPS 2020). However, given that four years have passed since its publication, does this work lack timeliness? Can its performance surpass current state-of-the-art methods?
2.	This paper uses a coreset-based replay method, so it should include more comparative experiments with other coreset selection approaches. The comparative experiments in this paper are insufficient.
3.	Since the paper mentions that reducible loss eliminates the need for backpropagation and reduces computational cost compared to traditional bi-level coreset selection strategies, an additional experiment should be included to validate this conclusion.
4.	Does training a holdout model using all the data contradict the continual learning (CL) setting?
5.	The paper mentions that “samples with high performance gain are both representative and informative.” Could the distribution of high-performance samples in feature space be provided as evidence for this claim?

---

> ### Author Response · Authors · 2024-11-21
> **Response**
>
> Thank you for your thoughtful and detailed feedback. We appreciate your recognition of the potential impact of our approach and your constructive suggestions for improving our work.
>
> **Weakness**
>
> 1. We have identified key hyperparameters in our ablation study (Section 5.3) and evaluated their impact. Additionally, we have addressed the design choice of removing coreset samples with low performance gain and included this discussion in Appendix Q. of our updated submission.
> 2. For a comprehensive comparison, we have included recent coreset selection methods for continual learning, such as GCR, OCS, and BCSR, in Table 1 of our updated submission. These results highlights the timeliness and effectiveness of our approach.
> 3. Furthermore, we have compared our method with Greedy Coreset and BCSR using more complex backbone models, specifically ResNet-50 for CIFAR-100 and VIT-Tiny for Tiny-ImageNet. The memory size was set to 1000 for CIFAR-100 and 2000 for Tiny-ImageNet respectively. The average accuracy and time cost are presented in Tables below. The results demonstrate both the effectiveness and efficiency of our method.
>
> Average accuracy and time cost on CIFAR-100
>
> |                | Average accuracy | Time cost |
> |----------------|:----------------:|:---------:|
> | Greedy Coreset | 28.17±0.57 | 21h 58m |
> | BCSR           | 27.32±0.10 | 11h 26m |
> | Our method     | 32.51±0.58 | 10h 04m |
>
> Average accuracy and time cost on Tiny-ImageNet
>
> |                | Average accuracy | Time cost |
> |----------------|:----------------:|:---------:|
> | Greedy Coreset | 12.17±0.07 | 24h 55m |
> | BCSR           | 12.33±0.08 | 28h 13m |
> | Our method     | 17.51±0.11 | 15h 13m |
>
> If you have any further concerns, please feel free to let us know.

---

> > ### Author Response · Authors · 2024-11-21
> > **Response to Questions**
> >
> > **Questions**
> >
> > 1. Coreset selection is a non-trivial, NP-hard problem, with Greedy Coreset being a seminal work and a widely cited baseline. In our introduction (line 46-63) and related work (line 97-103), we also discuss more recent bi-level coreset selection methods, including PBCS and BCSR. Our method specifically addresses the challenge of selecting non-representative samples for the coreset. PBCS also tends to select non-representative samples, as demonstrated in our ablation study on coreset selection from noisy data in Section 5.3. Similarly, the recent work BCSR, which employs implicit gradients to solve the bi-level optimization problem, may also select non-representative samples. In summary, our method tackles a unique and critical challenge in the field of coreset selection.
> > 2. We compared our method with recent coreset selection methods in Appendix L. We further include comparisons to BCSR, GCR, and OCS in Table 1 in the updated submission. The results demonstrate that our method consistently outperforms recent coreset selection approaches across all datasets, highlighting its effectiveness. Additionally, we have compared our method with Greedy Coreset and BCSR using more complex datasets and backbone models, as addressed in our response to Weaknesses.
> > 3. We have provided the time cost of coreset selection in Section 5.3 (line 511-524). As noted, Greedy Coreset requires over 4000 seconds to compute the Neural Tangent Kernel, resulting in a total coreset selection time exceeding 4000 seconds. In the experiment of large backbone models (in our response in weakness), our method costs less time compared to other bi-level coreset selection methods, further demonstrating its efficiency and effectiveness.
> > 4. In our CSRL-CL method, the holdout model is trained on the training dataset of the current task. In CSRL-CL-prv, the holdout model is trained on both the training dataset of the current task and the memory. Since the full previous data is unavailable and future task data is not incorporated, our approach does not violate the constraints of the continual learning setting.
> > 5. We have visualized the features of coreset samples selected by different methods from CIFAR-10 using t-SNE. The features were extracted using a ResNet-18 model trained on CIFAR-10, as shown in Figure 16 in our updated submission. Representative samples share common features with other samples and are effectively learned by the model, thus could be well classified. The features of the coreset selected by our method are concentrated near the high density part and better separated compared to those selected by the two baseline methods, demonstrating the ability of our approach to select representative samples.
> > &ensp;&ensp; Our work defines informative sample as samples contains new knowledge compared to currently selected coreset. To demonstrate that our method selects both representative and informative samples, we trained a model on the first 150 selected samples and visualized their features as dots. We then visualized the next 50 selected samples as triangles. Figure 17 shows the features of coreset samples selected by our method and by Greedy Coreset. The later-selected samples by our method tend to lie near the margins of feature clusters, indicating they are less well-learned by the model. Furthermore, samples selected by our method are more closely aligned with the feature clusters of each class, while those selected by Greedy Coreset are more scattered. This demonstrates that our method effectively selects samples that are both representative and informative. However, it is important to note that the informativeness of a sample cannot be determined solely based on its features, as not well learned samples, such as ambiguous or noisy ones, may negatively impact the model's learning process. Detailed discussion and figures are shown in Appendix T. in our updated submission.

---

> > > ### Comment · Reviewer_kUqK · 2024-11-26
> > >
> > > I appreciate the response of the authors. I have the following concerns:
> > >
> > > 1. Many coreset selection works in continual learning are not based on bi-level selection. Why does this work focus on bi-level optimization, what are the advantages?
> > > 2. The comparison experiment should involve some SOTAs which are not based on bi-level selection.
> > > 3. The tsne experiment cannot explain why the selected samples are representative and informative. And the tsne should be conducted on some works without bi-level selection strategy. If the visualization only shows the aggregation of the selected samples, then why not use a prototype-based method that selects samples near the class prototypes.

---

> > > > ### Author Response · Authors · 2024-11-27
> > > > **Response to further concerns**
> > > >
> > > > Thank you for your thoughtful follow-up and for engaging with our rebuttal. We appreciate your detailed questions and the opportunity to further clarify our work. Below, we address your concerns:
> > > >
> > > > 1. A coreset is a subset of the full dataset designed so that a model trained on the coreset achieves performance comparable to one trained on the full dataset. In rehearsal-based continual learning, the coreset serves as an informative subset for replay, enabling the model to retain knowledge of previous tasks effectively.
> > > >
> > > >     Our focus on bi-level optimization stems from its ability to directly optimize the selection process for maximizing downstream task performance, rather than relying on heuristics. Unlike other coreset selection methods, bi-level approaches explicitly evaluate how changes to the coreset impact the performance of a model trained on it, considering both the selection process and subsequent retraining.
> > > >
> > > >     As demonstrated in Appendix B, our CSRL method approximates the implicit gradient while addressing the challenge of selecting non-representative samples. Additionally, CSRL is more computationally efficient compared to traditional bi-level optimization methods.
> > > > 2. We have included coreset selection methods OCS and GCR in Table 1. These methods use gradient matching for coreset selection and are not based on bi-level optimization. The results demonstrate that our method outperforms both OCS and GCR across all datasets.
> > > > 3. We agree that the t-SNE feature map does not explicitly demonstrate why the selected samples are representative and informative. Our method selects samples that are representative and contain new information compared to the already selected subset. However, the feature map cannot indicate whether adding a specific sample to the coreset and retraining the model on the updated coreset would result in maximum performance improvement. In other words, the feature map in Figures 16 and 17 represents the outcome of the selection process, not the reasoning behind it.
> > > >
> > > >     In our updated submission, we have included t-SNE feature maps of selected samples from iCaRL's selection and k-means feature selection in Figure 18 (Appendix T). iCaRL’s selection, as shown in Table 1, uses a herding method that selects samples near the mean of features for each class and ensures that the class feature means are preserved. The results in Table 1 highlight the effectiveness of our method compared to iCaRL’s selection.
> > > >
> > > >     Selecting samples near the class feature mean can lead to redundant information, as samples close in feature space often contain similar information. In contrast, our method selects samples that are not well-represented by the current coreset. Redundant samples will have lower CSRL scores and are therefore not selected.

---

> ### Comment · Reviewer_kUqK · 2024-11-27
>
> 1. Some coreset selection methods such as [1,2,3] are all without bi-level strategy, are they also just heuristic? Also, they are SOTAs of coreset selection methods, maybe more comparison experiments are supposed to be conducted with them.
> 2. You claim that “samples with high performance gain are both representative and informative” in the abstruct, why works like [4,5] can not be considered as comparisons? They also make sample selections or updates based on the performance gain.
>
> [1] Yang S, Cao Z, Guo S, et al. Mind the Boundary: Coreset Selection via Reconstructing the Decision Boundary. Forty-first International Conference on Machine Learning. 2024.
>
> [2] Wan Z, Wang Z, Wang Y, et al. Contributing Dimension Structure of Deep Feature for Coreset Selection. Proceedings of the AAAI Conference on Artificial Intelligence. 2024, 38(8): 9080-9088.
>
> [3] Xu X, Zhang J, Liu F, et al. Efficient adversarial contrastive learning via robustness-aware coreset selection. Advances in Neural Information Processing Systems, 2024, 36.
>
> [4] Aljundi R, Belilovsky E, Tuytelaars T, et al. Online continual learning with maximal interfered retrieval. Advances in neural information processing systems, 2019, 32.
>
> [5] Jin X, Sadhu A, Du J, et al. Gradient-based editing of memory examples for online task-free continual learning. Advances in Neural Information Processing Systems, 2021, 34: 29193-29205.

---

> > ### Author Response · Authors · 2024-11-28
> > **Response to further qestions**
> >
> > Thank you for your detailed follow-up and for raising these important points. Below, we address your concerns:
> >
> > 1. We conducted coreset selection experiments with [2] and [3] on the CIFAR-10 and CIFAR-100 datasets under the same settings as Section 5.1, with the coreset size set to 200. The results, presented in Table below, demonstrate the effectiveness and timeliness of our method. Additionally, we note that RCS [2] is specifically designed for adversarial contrastive learning, which introduces a domain mismatch with our work, leading to its lower performance. The source code or detailed implementation of [1] is not publicly available, which makes it challenging to reproduce the method and perform a direct comparison under the same experimental settings. We remain open to including [1] in future evaluations if its implementation becomes available. We have included these additional results in Appendix U in our updated submission.
> >
> > | Method         | CIFAR-10 | CIFAR-100 |
> > |----------------|:--------:|:---------:|
> > | RCS            | 27.17±1.11 | 4.87±0.60 |
> > | GC+CDS         | 33.03±1.74 | 8.06±0.23 |
> > | Greedy Coreset | 36.53±1.40 | 6.94±0.23 |
> > | PBCS           | 37.41±0.26 | 6.77±0.18 |
> > | CSRL (ours)    | 39.58±0.14 | 8.91±0.42 |
> >
> > 2. Our method addresses the task of selecting samples from the current training dataset for inclusion in memory, with discarded samples no longer being used in future steps. In contrast, MIR [4] focuses on selecting highly interfered samples from memory for replay, allowing discarded samples to be reconsidered in later steps. GMED [5], on the other hand, edits memory samples to maximize interference on memory data. Since our method tackles a different sub-task, we did not directly compare it with these baselines.\
> > However, our method can be integrated with these approaches by replacing reservoir sampling in their pipelines with the CSRL-RL strategy proposed in Section 4.3. We have already demonstrated the effectiveness of our method by combining it with strong baselines such as DER++ and LODE. Due to time constraints, we will provide results of integrating our method with MIR and GMED in later comments.

---

> > > ### Author Response · Authors · 2024-12-01
> > > **Further continual learning results**
> > >
> > > 1. We have integrated our CSRL-RS method, proposed in Section 4.3, with MIR and GMED and evaluated these combinations under the same settings as described in Section 5.2.2.
> > > 2. The results, presented in the tables below, demonstrate that incorporating our method consistently improves performance. This highlights our method's ability to select more informative subsets for memory and further emphasizes its compatibility with existing experience replay (ER) methods.
> > >
> > > Average accuracy on CIFAR-100 dataset.
> > >
> > > |           | 200 memory | 500 memory |
> > > |-----------|:----------:|:----------:|
> > > | MIR       | 13.71±0.80 | 21.75±0.80 |
> > > | CSRL-MIR  | 15.45±0.34 | 22.59±0.45 |
> > > | GMED      | 13.44±0.30 | 20.44±0.77 |
> > > | CSRL-GMED | 14.62±0.13 | 21.98±0.34 |
> > >
> > > Average accuracy on Tiny-ImageNet dataset.
> > >
> > > |           | 200 memory | 500 memory |
> > > |-----------|:----------:|:----------:|
> > > | MIR       | 8.56±0.11  | 10.31±0.18 |
> > > | CSRL-MIR  | 8.71±0.16  | 10.82±0.05 |
> > > | GMED      | 8.34±0.20  | 9.93±0.18  |
> > > | CSRL-GMED | 8.61±0.30  | 10.58±0.19 |
> > >
> > > We hope these experiments could address your concern.

---

> > > > ### Author Response · Authors · 2024-12-02
> > > > **We hope our responses could address your concerns**
> > > >
> > > > Dear Reviewer kUqK,
> > > >
> > > > Thank you for your comments and for engaging with the discussion phase. We hope that our rebuttal and the additional analyses have adequately addressed your concerns, including timeliness and effectiveness.
> > > >
> > > > If our responses have resolved your concerns, we kindly ask that you consider updating your score to reflect the current evaluation. If there are any remaining clarifications or follow-up questions, we would be happy to address them promptly.
> > > >
> > > > We sincerely appreciate the time and effort you’ve dedicated to reviewing our submission and providing valuable feedback to improve our work.
> > > >
> > > > Best regards,
> > > > Authors of Submission 1231

---

> > > > > ### Comment · Reviewer_kUqK · 2024-12-03
> > > > >
> > > > > Based on the reply, I raise my score, but the necessity of bi-level optimization and the comparison with sotas need to be explained clearly in the paper.

---

> > > > > > ### Author Response · Authors · 2024-12-03
> > > > > > **Thank you for your feedback**
> > > > > >
> > > > > > Dear Reviewer kUqK,
> > > > > >
> > > > > > Thank you for your thoughtful engagement during the discussion phase and for raising your score. We sincerely appreciate your recognition of our efforts to address your concerns.
> > > > > >
> > > > > > We have added comparisons to other SOTA methods in Appendix U due to space limitations. We also acknowledge the importance of clearly explaining the necessity of bi-level optimization. We will ensure that the necessity of bi-level coreset selection and the comparisons with SOTAs are more thoroughly detailed to enhance the clarity and impact of our work.
> > > > > >
> > > > > > Thank you again for your constructive feedback, which has been invaluable in improving our paper.
> > > > > >
> > > > > > Best regards,
> > > > > > Authors of Submission 1231

---

### Official Review · Reviewer_MUYB · 2024-11-03

**Soundness:** 4
**Presentation:** 4
**Contribution:** 4
**Rating:** 8
**Confidence:** 3

**Summary:**

The work presents a concise way of preventing select noisy data into the coreset due to their high loss value. By using a holdout model as the baseline, it incrementally selects samples with high log prob loss. Which is mathematically shown by the authors that such samples are informative but not well represented yet in the coreset. The work shows improvements on various common continual learning datasets.

**Strengths:**

1. A detailed, structured, and rigorous paper with extensive derivations and experiments.
2. Conducts a variety of experiments, including those on standard continual learning datasets, more challenging ImageNet variants, training overhead, and extensive ablation studies.
3. Presents a clear and straightforward idea for selecting good samples to add to the coreset, supported by sound equations.

**Weaknesses:**

1. Not easy to understand at the first glance. It might be better to add some teaser figure.

**Questions:**

1. How do you train the holdout model? I believe it is trained only on the initial dataset, without any information about the incoming task. What if the holdout model is not good enough to calculate a meaningful loss for later tasks? For example, if the incoming task is very different from the previous one (a huge domain shift), the loss of holdout model might be poor for every input. In such a scenario, CSRL might not work.
2. What's the difference between eq.4 and eq.7? What's the difference between $\mathcal{S}$ and $\mathcal{M}$?

---

> ### Author Response · Authors · 2024-11-21
> **Response**
>
> Thank you for your detailed review, thoughtful questions, and constructive suggestions. We appreciate your recognition of the strengths of our work and the valuable feedback regarding clarity and the holdout model's adaptability. Your insights have helped us identify areas for improvement, and we’ve addressed your comments thoroughly in our responses. We are grateful for your time and effort in evaluating our submission.
>
> **weakness**
>
> We have included an overview figure of our CSRL-CL method in Appendix P. in the updated version of our paper for a clear understanding. If you have any further concerns, please feel free to let us know.
>
> **Questions**
>
> 1. In the coreset selection task, we train a holdout model on the entire dataset until convergence. For the continual learning task, we treat the training dataset of the current task as the entire dataset and select the coreset from it. After training the continual learning (CL) model on each task, the holdout model is then trained on the dataset of the current task. In CSRL-CL-prv (described in Section 4.2), holdout model is trained on current task dataset and memory. The holdout model is updated after training on each task, rather than being trained only at the initial step. As a result, holdout model incorporates updated information, enabling it to provide valuable insights for the selection process.
> Since the future data distribution in continual learning is unknown, our goal is to extract an informative subset from the seen datasets. This subset is replayed during training on future tasks to mitigate forgetting of previous tasks. Therefore, the holdout model only needs to provide information relevant to the seen tasks and does not need to account for unseen tasks. Figure 14 in our updated submission illustrates the coreset selection process for continual learning.
> 2. Equation (4) and Equation (7) represent the same equation applied to different tasks. Equation (4) pertains to coreset selection tasks, where $\mathcal{S}$ represents the currently selected subset and $(\mathbf{x},\mathbf{y})$ denotes the full dataset. While Equation (7) is used in the context of continual learning, where the subscript $t$ indicates the task ID. $\mathcal{S_t}$ represents the subset selected from the training dataset of task $t$, and $(\mathbf{x_{1:t}},\mathbf{y_{1:t}})$ denotes the data available from all seen tasks, specifically the current task's training dataset $\mathcal{D_t}$ and the memory $\mathcal{M}$.
> &ensp;&ensp; In coreset selection task, $\mathcal{S}$ denotes the subset, while in continual learning, $\mathcal{M}$ refers to the memory.  In continual learning, coresets are selected from each task, and the memory $\mathcal{M}$ contains coresets from all previously seen tasks. Therefore, $\mathcal{M}=\cup_{i}^{t}\mathcal{S}_i$.

---

> > ### Comment · Reviewer_MUYB · 2024-11-25
> >
> > Thanks for clarifying, I have no further concerns.

---

> > > ### Author Response · Authors · 2024-11-25
> > > **Thank you for your follow-up**
> > >
> > > Dear Reviewer MUYB,
> > >
> > > Thank you for your follow-up and for taking the time to review our clarifications. We sincerely appreciate your thoughtful feedback and your efforts in evaluating our work.
> > >
> > > If any additional questions or concerns arise during the remaining discussion phase, we would be happy to address them. Thank you again for your support and valuable insights throughout this process.
> > >
> > > Best regards,\
> > > Authors of submission 1231.

---

### Author Response · Authors · 2024-11-21
**General response**

Dear Reviewers,

We sincerely appreciate the time and effort you have dedicated to reviewing our submission. Your thoughtful questions, constructive suggestions, and detailed feedback have been invaluable in helping us refine and improve our work.

We have carefully reviewed all the comments and addressed the key concerns raised. We provided additional clarifications on the methodology, expanded experimental comparisons, and included new results to further support the contributions of our approach. Your insights have also guided us in identifying areas for potential future improvements, which we acknowledge in our responses and revisions.

We are committed to engaging actively in the discussion phase to ensure that all concerns are addressed thoroughly. Once again, thank you for your thoughtful evaluations and for helping us strengthen this work.


**Contribution**

"Presents a clear and straightforward idea for selecting good samples to add to the coreset, supported by sound equations." --Reviewer MUYB

"By focusing on performance gain, this approach reduces computational demands and mitigates the impact of noisy data. It addresses key continual learning challenges, such as task interference and data streaming, making it particularly valuable for dynamic and large-scale applications." --Review kUqK

**Writing**

"A detailed, structured, and rigorous paper with extensive derivations and experiments." --Reviewer MUYB

"The paper is well-organized and writing is almost clear." --Reviewer Ungg

**Experiments**

"rigorous paper with extensive derivations and experiments." --Reviewer MUYB

"I found multiple supplementary explanations for the main text in the supplementary materials, which I believe is commendable and should be encouraged." --Reviwer Ungg

**Concerns**

Reviewer MUYB raised concerns about the clarity of our work. In response, we have provided additional clarifications.

Reviewer kUqK raised key concerns regarding timeliness, comprehensiveness, and clarity. In response, we have provided the necessary experiments, analyses, and clarifications to address these issues.

Reviewer Ungg raised major concerns regarding the soundness and comprehensiveness of our work. In response, we have provided detailed analysis and clarifications to address soundness and conducted additional experiments to enhance the comprehensiveness of our work.

We have provided detailed responses to all questions from each individual reviewer and updated our submission, using orange highlights to indicate the changes.

---

> ### Author Response · Authors · 2024-11-25
> **Follow-Up on Rebuttal Submission**
>
> Dear Reviewers,
>
> We hope this message finds you well. We are writing to kindly follow up on the rebuttal we submitted a few days ago. We have provided detailed responses to your comments and would be happy to clarify or expand on any points if needed.
>
> We understand that everyone is busy, and we greatly appreciate the time and effort you are dedicating to reviewing our submission. Please let us know if there are additional questions or feedback we can address during this discussion phase.
>
> Thank you again for your thoughtful evaluations and guidance in improving our work.
>
> Best regards,\
> Authors of submission 1231.

---

### Meta-Review · Area_Chair_SUTw · 2024-12-23

**Metareview:**

The paper introduces a coreset selection method for continual learning, using reducible loss to efficiently select representative samples and improve model performance. Reviewers feel positive about this paper due to its detailed derivations, extensive experiments on various datasets, and rigorous ablation studies, capable of addressing challenges like task interference and data streaming. In the meantime, reviewers raised several major concerns, including the usage and motivation of bi-level coreset selection strategies, the effectiveness against current methods, and the validity of the experimental comparisons. The authors successfully addressed these concerns in the discussion, and the AC was convinced.

**Additional Comments On Reviewer Discussion:**

There are extensive discussions between reviewers and authors. One reviewer is satisfied with the authors' responses and raises the rating, while the other seems not and maintains the rating. The authors did a good job in their response and provided adequate evidence and explanations, which are convincing to the AC.

---

### Decision · Program_Chairs · 2025-01-22

Accept (Poster)